# Unsupervised Behavioral Tokenization and Action Quantization via Maximum Entropy Mixture Policies with Minimum Entropy Components

## Abstract

A fundamental problem in reinforcement learning is how to learn a concise discrete set of behaviors that can be easily composed to solve any downstream task. An effective "tokenization" of behavior requires tokens to be transferable, learnable in an unsupervised manner, and short-lived to facilitate composability, akin to tokens in large-language models. Previous work has studied this problem using action quantization, sub-policies, options and skills, but no solution provides an unsupervised, task-agnostic method to discover short-lived, action-spanning behavioral tokens. We introduce an online unsupervised framework that autonomously discovers behavioral tokens via a joint entropy objective—maximizing the cumulative entropy of an overall mixture policy to ensure rich, exploratory behavior under the maximum occupancy principle, while minimizing the entropy of each component policy to enforce diversity and high specialization. We prove convergence in the tabular setting of our policy iteration algorithm, then extend to continuous control by fixing the discovered components and deploying them deterministically as quantized actions within an online optimizer to maximize reward. Experiments demonstrate that our *maxi-mix-mini-com* entropy-based tokenization and action quantization provide interpretable and reusable behavioral tokens that can be sequenced to achieve competitive performance against continuous-action controllers and that largely outperform random action quantization and skill discovery methods.

## 1 Introduction

Continuous control tasks in biophysical systems and robots require mapping state inputs to high-dimensional continuous actions in real time (Wolpert et al., 1994). This presents significant challenges: the vastness of the continuous action space makes exploration inefficient (Lillicrap et al., 2015), and balancing the exploration–exploitation trade-off often necessitates approximations that complicate policy optimization (Dalal et al., 2021). One way to alleviate this problem is to use a concise set of *behavioral tokens* that are task-agnostic, reusable, transferable, and can be easily and flexibly composed to solve any downstream task. By focusing on core representative tokens, behavioral tokenization can improve sample efficiency, accelerate convergence, and avoid wasteful exploration of irrelevant continuous actions (Dadashi et al., 2021). Beyond its technical strengths, behavioral tokenization can allow for simpler labeling and interpretability by human observers, making agent behavior easy to understand. It can also simplify planning or sequence generation by arbitrarily composing tokens (Chebotar et al., 2023; Sheebaelhamd et al., 2025).

Previous work has used action quantization (AQ), sub-policies, options, and skills as ways to build token-like behaviors. However, these fail in one or more fundamental aspects, making them non-transferable, not learned in an unsupervised manner, long-lived and difficult to compose, or not action-centered. In AQ methods (Luo et al., 2023; Seo et al., 2024), the aim is to partition the continuous action space into a concise discrete, state-dependent set of actions. Aside from naive per-dimension action binning, which suffers from the curse of dimensionality (Chebotar et al., 2023), current successful methods rely on curated offline datasets (Luo et al., 2023) or expert demonstrations (Dadashi et al., 2021) to build the action set. However, this approach ties the discovered quantized

actions to the dataset's task, hindering both generalization and reusability of the behavioral tokens (Luo et al., 2023). Another direction to behavioral tokenization, which is closest to ours, is to express the policy as a mixture of component sub-policies (Daniel et al., 2016; Seyde et al., 2022). This leads to short-lived (one step) behavioral tokens, but current approaches are task-dependent, making tokens not transferable (Daniel et al., 2016). Further, in existing algorithms, the learned policy collapses to a single component as learning progresses (e.g., Sheebaelhamd et al. (2025)).

Options and skills methods discover behavioral tokens that are time-extended and state-centric (Eysenbach et al., 2019; Bacon et al., 2017). Although promising, these methods have two major limitations regarding effective behavioral tokenization. First, long-lived behavioral tokens are more difficult to compose than short-lived ones. Long-lived tokens—such as options or skills—must execute for many consecutive timesteps before a new token can be selected, which limits the agent's ability to rapidly interleave different behaviors in response to changing state conditions. Short-lived tokens, by contrast, can be freely recombined at every timestep, enabling fine-grained behavioral composition. Here, we draw an analogy to large language models, which operate on sequences of discrete sub-word tokens (Vaswani et al., 2017)—rather than sentence-level primitives—because fine-grained units enable compositional generalization that sentence-level atoms cannot. Second, an *action-centric* view of behavioral tokenization is more powerful than a *state-centric* one. In a state-centric view, behavioral tokens are primarily associated to *where* an agent goes (state marginals, terminal states, or trajectories). For instance, existing skill discovery methods (Eysenbach et al., 2019; Park et al., 2022; 2024; Sharma et al., 2020) are state-centric: they couple a latent variable to states or state trajectories by maximizing their mutual information. In contrast, in an action-centric view, diverse behavioral tokens are coupled to the *same* state (e.g, being able to walk and run in different directions from the same location), giving more flexibility to behavioral composability. Therefore, a *bank* of behavioral tokens must satisfy two criteria to be useful: (1) effective action spanning, so that the most representative local ways of acting at every state are sufficiently captured to synthesize diverse downstream tasks by composition; and (2) distinctness, so that tokens are efficiently searchable by a higher-level controller.

Here, we address two challenges: how to discover in an unsupervised manner state-dependent behavioral tokens that are short-lived and action-centric, and how to compose them to solve any downstream task. Using Markov Decision Processes (MDPs) in discrete time, we define behavioral tokens as one-step component policies of a mixture policy. We frame action behavioral tokenization as the problem of learning a finite set of component policies each with minimal action entropy, whose combination into a mixture maximizes cumulative action entropy (Fig. 1a). This *maxi-mix-mini-com entropy* formulation for behavioral tokenization is entirely unsupervised and does not rely on any offline dataset, task-specific definition or demonstrations. The goal of maximizing cumulative action entropy—the maximum occupancy principle (MOP) (Ramírez-Ruiz et al., 2024; Moreno-Bote & Ramirez-Ruiz, 2023)—promotes producing all behaviors compatible with the agent's dynamics and environment, while the goal of minimizing components' entropy enforces diversity and distinctness among tokens (Fig. 1b). The result is a bank of distinct tokens that effectively spans action state and can be easily composed to solve any downstream task (Fig. 1c). Because action bottlenecks or terminal states (e.g., damage to the agent) reduce action entropy, learned tokens produce span-seeking and safe actions that try to avoid them (Fig. 1a); tokens are reusable across tasks and environments as they are optimized to jointly generate cumulative action entropy, rather than solving any task.

We first show that the maxi-mix-mini-com entropy objective is tractable, and we provide a provably convergent policy-iteration algorithm for the discrete setting (Sec. 2). Our algorithm simultaneously learns the optimal component policies and mixture weights, and both are generally state-dependent. Building on the tabular case, we extend our theory to the continuous action case (Sec. 3), providing the first unsupervised online AQ method, where quantized actions are the modes of the discovered components. We empirically show that learned quantized actions scale effectively, transferring successfully to novel environment layouts and achieving performance close to leading continuous controllers such as SAC (Haarnoja et al., 2018) and PPO (Schulman et al., 2017) in high-dimensional, on-policy control tasks (Sec. 4) with as few as $\sim 16$ components. Both skill discovery methods and naive random AQ systematically underperform our approach. Our results establish a principled framework for behavioral tokenization and AQ resulting in interpretable components that can be flexibly composed for efficient exploration and generalize well to unseen tasks.

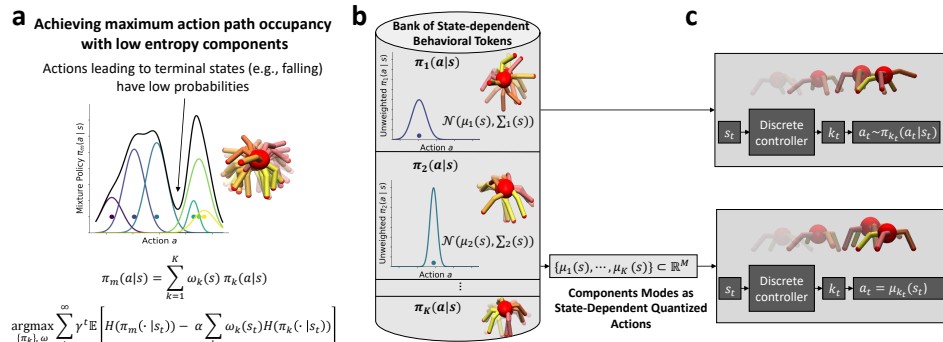

Figure 1: Behavioral token discovery via unsupervised *maxi-mix-mini-com* entropy learning. **(a)** The components of a mixture policy are learned to maximize cumulative action entropy under MOP while minimizing individual entropy (bottom), which entails avoiding falling close to action bottlenecks or terminal states (top). **(b)** The result is a bank of $K$ distinct behavioral tokens (components $\pi_k$). In continuous action spaces, we take components to be Gaussian, so learning delivers means $\mu_k(s)$ and diagonal covariance matrices $\Sigma_k(s)$. **(c)** For downstream tasks, these components are frozen and treated as black-box tokens for any discrete-action algorithm (e.g., DQN). The discrete action space is the index set $\{1, \ldots, K\}$, where selecting index $k$ executes component $\pi_k$ at state $s$ either by sampling an action from $\pi_k(\cdot \mid s)$ or by using its mode $\mu_k(s)$ to accumulate task-specific reward.

## 2 MAXI-MIX-MINI-COM ENTROPY OBJECTIVE FOR TOKENIZATION

We aim to learn task-agnostic, short-lived, state-dependent behavioral tokens that facilitate generation of diverse behaviors and the solution of downstream tasks. We first start with the discrete action-state case, and later address the continuous action-state case (Sec. 3). We consider a MDP $(\mathcal{S}, \mathcal{A}, p, \gamma)$, where $\mathcal{S}$ is a discrete state space, $\mathcal{A}(s)$ is a state-dependent discrete action space, $p(s'|s, a)$ is the transition kernel, and $\gamma \in [0, 1)$ is the discount factor. We define the mixture policy

$$\pi_m(a|s) = \sum_{k=1}^{K} w_k(s)\pi_k(a|s) \tag{1}$$

as a non-negative linear combination of $K$ component policies $\pi_k(a|s)$ with mixture weights $w_k(s) \geq 0$ and $\sum_k w_k(s) = 1$. Weights are state-dependent, i.e., they are not fixed, and they can be interpreted as the probability that policy $k$ is chosen at state $s$. The number of components $K$ can be made state-dependent, $K(s)$, in a straightforward manner.

Our goal is to optimize both the component policies and weights so as to maximize the value function

$$V(s) = \sum_{t=0}^{\infty} \gamma^t \mathbb{E}\left[\mathcal{H}(\pi_m(\cdot|s_t)) - \alpha \sum_k w_k(s_t)\mathcal{H}(\pi_k(\cdot|s_t))\Big| S_0 = s\right], \tag{2}$$

where the expectation is over actions sampled from the mixture policy and state transitions given initial condition $s$, $\mathcal{H}(\pi(\cdot|s)) = -\sum_a \pi(a|s) \log \pi(a|s)$ is the action entropy of a policy $\pi$ given state $s$, and $0 \leq \alpha < 1$. This goal maximizes the sum of future action entropies, thus generating diversity of action paths, while minimizing the entropy of each of the components, thus favoring specialized component policies. These two generally opposing goals are balanced by the hyperparameter $\alpha$. The degree of produced diversity is also shaped by the state-dependence of the action set $\mathcal{A}(s)$. Indeed, most realistic environments have action bottlenecks (e.g., corridors) and terminal states (e.g., zero battery energy) where the size of available actions is reduced to a few or just one action (e.g., not doing anything when dead or when having zero energy). State-dependence actions sets strongly shape the solution, leading to maximum entropy policies that are non-uniform, promoting exploration and diverse behavior while trying to avoid bottlenecks and terminal states (Ramírez-Ruiz et al., 2024).

We note that replacing the weighted sum of component policy entropies in Eq. 2, $-\sum_k w_k(s_t)\mathcal{H}(\pi_k(\cdot|s_t))$, by an unweighted one, $-\sum_k \mathcal{H}(\pi_k(\cdot|s_t))$ would penalize as well having

entropic component policies, but it would not lead to an easily solvable problem (see Appendix A.2.1). Importantly, when fixing the mixture policy in Eq. 2, a solution having distinct active components has a higher value than a solution with identical or inactive ones (see last remark in Appendix A.2 ). Thus, the optimal solution never collapses to a trivial solution.

Eq. 2 can be recursively expressed in the form of the Bellman equation

$$V(s) = \mathcal{H}(\pi_m(\cdot|s)) - \alpha \sum_k w_k(s)\mathcal{H}(\pi_k(\cdot|s)) + \gamma \sum_a \pi_m(a|s)Q(s,a) \tag{3}$$

$$Q(s,a) \equiv \sum_s p(s'|s,a)V(s') \,, \tag{4}$$

where $Q(s,a)$ is the expected continuation action-value function from the successor state – not to be confused with the conventional $Q$-action-value function, which also includes immediate reward. Therefore, the objective can be expressed as maximizing the value function over components and mixture weights through the optimality Bellman equation

$$V^*(s) = \max_{\pi,w} \left( R(s;\pi,w) + \gamma \sum_a \pi_m(a|s)Q^*(s,a) \right) \tag{5}$$

$$R(s;\pi,w) \equiv \mathcal{H}(\pi_m(\cdot|s)) - \alpha \sum_k w_k(s)\mathcal{H}(\pi_k(\cdot|s)) \,, \tag{6}$$

where $\pi$ indicates the set of component policies $\pi = \{\pi_1, ..., \pi_K\}$, $w = \{w_1, ..., w_K\}$, and the conditioning on the state $s$ is omitted but understood.

We first prove that for a high capacity mixture policy (i.e., when the number of components is greater than or equal to the size of the action space) solving Eq. 5 is equivalent to solving an unconstrained problem with no restrictions on the mixture policy. Although straightforward, this result shows that there is a limit where the problem converges to a known solution.

**Theorem 1.** *If $K(s) \geq |\mathcal{A}(s)|$ for all $s$, then the optimal solution of Eq. 5 is such that $\pi_k(a_k|s) = 1$ only for one action $a_k$ and the set of actions spans $\mathcal{A}(s)$, that is, $\{a_k\} = \mathcal{A}(s)$. Then, $\pi_m$ in Eq. 1 is unconstrained and the optimal value function obeys*

$$V^*(s) = \max_{\pi_m} \left( \mathcal{H}(\pi_m(\cdot|s)) + \gamma \sum_a \pi_m(a|s)Q^*(s,a) \right) \,. \tag{7}$$

*The optimal mixture policy is unique, while the mixture components are not unique if $|\mathcal{A}(s)| > 1$ at least for one state.*

*Proof.* See Appendix Sec. A.2.

While the case $K(s) \geq |\mathcal{A}(s)|$ is interesting, the most relevant scenario occurs when the number of components is smaller than the action dimensionality, $K(s) < |\mathcal{A}(s)|$, as this will lead to a compression and quantization of action space into components that can later be used to compose sequences of actions for reward maximization. If mixture capacity is low, directly optimizing Eq. 5 is hard because derivatives with respect to both $\pi$ and $w$ lead to a set of nonlinear equations that to our knowledge cannot be solved explicitly (Sec. A.2). To bypass this problem, we derive an iterative algorithm, which is the main theoretical contribution of our paper.

We first introduce the responsibility

$$r_k^*(s,a) = \frac{w_k(s)\,\pi_k(a|s)}{\pi_m(a|s)} \,, \tag{8}$$

which is the probability that a performed action $a$ at state $s$ has been generated from the component policy $k$ – note that the denominator, defined in Eq. 1, makes $r_k^*(s,a)$ probabilities for all $(s,a)$, that is, $r_k^*(s,a) \geq 0$ and $\sum_k r_k^*(s,a) = 1$. Next, we define the immediate gain $G(s;\pi,w,r)$ as

$$G(s;\pi,w,r) = -\sum_k w_k(s)\log w_k(s) - (1-\alpha)\sum_{k,a} w_k(s)\,\pi_k(a|s)\log \pi_k(a|s)$$

$$+ \sum_{k,a} w_k(s)\pi_k(a|s)\log r_k(s,a) \,, \tag{9}$$

where $r = \{r_1, ..., r_K\}$ is a probability vector. It can be easily seen that (i) $G(s; \pi, w, r^*) = R(s; \pi, w)$ when using the probability vector $r^*$ given by Eq. 8, and (ii) $G(s; \pi, w, r^*) \geq G(s; \pi, w, r)$ for any probability vector $r$ (Arimoto, 1972). The above definitions and observations allow us to write the optimality Bellman equation 5 as

$$V^*(s) = \max_{\pi, w} \max_{r} \left( G(s; \pi, w, r) + \gamma \sum_a \pi_m(a|s) Q^*(s, a) \right) . \qquad (10)$$

As in the Blahut–Arimoto algorithm (Blahut, 1972; Arimoto, 1972) and in more recent applications (Daniel et al., 2016; Arenz et al., 2020), we use this fact to build an algorithm showing monotonic improvement of the value function until convergence.

Our algorithm, described in Algorithm 1, proceeds as follows:

(1) Initial conditions: Start the value function $V^{(0)}(s) = 0$ for all $s$ and arbitrary $\pi^{(0)} = \{\pi_1^{(0)}, ..., \pi_K^{(0)}\}$ and $w^{(0)} = \{w_1^{(0)}, ..., w_K^{(0)}\}$ outside the simplex boundaries, that is, $\pi_k^{(0)}(a|s) > 0$ and $w_k^{(0)}(s) > 0$ for all $s$ and $a$. Define the initial responsibilities as

$$r_k^{(0)}(s, a) = w_k^{(0)}(s) \, \pi_k^{(0)}(a|s) \big/ \pi_m^{(0)}(a|s) > 0 . \qquad (11)$$

Iterate the following step (2) for $n = 1, 2, ....$

(2) At iteration $n$, start from $V^{(n-1)}$, $\pi^{(n-1)}$, $w^{(n-1)}$ and $r^{(n-1)}$ and perform steps (2a)-(2d):

(2a) Define the updated value function $V^{(n)}$ as

$$V^{(n)}(s) = G(s; \pi^{(n-1)}, w^{(n-1)}, r^{(n-1)}) + \gamma \sum_a \pi_m^{(n-1)}(a|s) Q^{(n-1)}(s, a) , \qquad (12)$$

with $Q^{(n-1)}(s, a) = \sum_{s'} p(s'|s, a) V^{(n-1)}(s')$ and $\pi_m^{(n-1)}(a|s) = \sum_k w_k^{(n-1)}(s) \pi_k^{(n-1)}(a|s)$.

(2b) Compute $\pi^{(n)} = \{\pi_1^{(n)}, \cdots, \pi_K^{(n)}\}$ fixing $V^{(n-1)}$ and $r^{(n-1)}$ using

$$\pi_k^{(n)}(a|s) = \frac{[r_k^{(n-1)}(s, a)]^{\frac{1}{1-\alpha}} e^{\frac{\gamma}{1-\alpha} Q^{(n-1)}(s, a)}}{\sum_{a'} [r_k^{(n-1)}(s, a')]^{\frac{1}{1-\alpha}} e^{\frac{\gamma}{1-\alpha} Q^{(n-1)}(s, a')}} . \qquad (13)$$

(2c) Compute $w^{(n)}$ fixing $V^{(n-1)}$ and $r^{(n-1)}$ as

$$w_k^{(n)}(s) = \frac{\left( \sum_a [r_k^{(n-1)}(s, a)]^{\frac{1}{1-\alpha}} e^{\frac{\gamma}{1-\alpha} Q^{(n-1)}(s, a)} \right)^{1-\alpha}}{\sum_{k'} \left( \sum_a [r_{k'}^{(n-1)}(s, a)]^{\frac{1}{1-\alpha}} e^{\frac{\gamma}{1-\alpha} Q^{(n-1)}(s, a)} \right)^{1-\alpha}} . \qquad (14)$$

(2d) Compute $r^{(n)}$ fixing $V^{(n-1)}$, $\pi^{(n)}$ and $w^{(n)}$ using

$$r_k^{(n)}(s, a) = w_k^{(n)}(s) \, \pi_k^{(n)}(a|s) \big/ \pi_m^{(n)}(a|s) , \qquad (15)$$

with $\pi_m^{(n)}(a|s) = \sum_k w_k^{(n)}(s) \, \pi_k^{(n)}(a|s)$.

**Theorem 2.** *The algorithm following steps (1) and (2), which consists in iterating the $\pi$, $w$ and $r$ using Eqs. 12,13,14,15 in this order, improves the value function monotonically until convergence.*

*Proof.* See Appendix Sec. A.2.

### 2.1 EXPERIMENTS AND RESULTS

We first validate Algorithm 1 in a deterministic grid-world with impassable walls and terminal states (see Fig. 4 in Appendix). The action set is {up, right, down, left} when away from the walls. Any action that would move into a wall is not available. When capacity is large, $K = 4 = \max_s |\mathcal{A}(s)|$, we find that component policies are sharply peaked at a single action, and they span all actions (Fig. 2a), as predicted by Theorem 1. When capacity is smaller (e.g., $K = 3$), some components become sharp, while others are not (Fig. 2b), but all of them together offer diversity

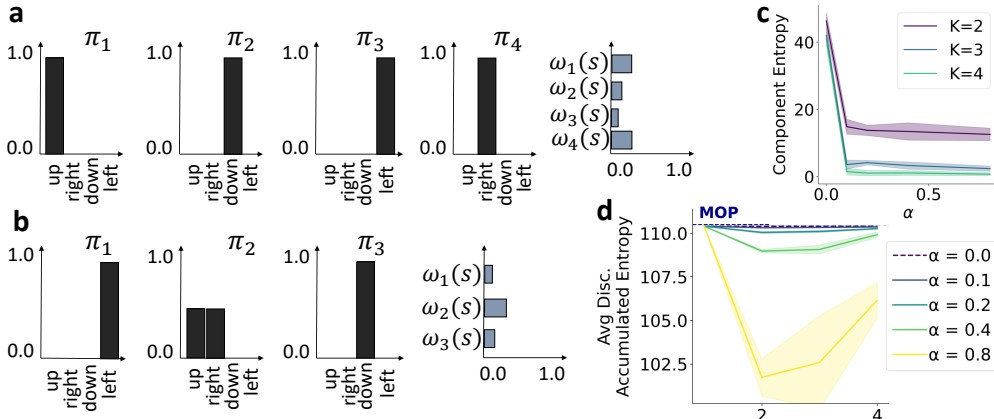

Figure 2: Component specialization and cumulative mixture entropy in a grid world (Fig. 4, Appendix A.5). **(a,b)** Learned state-dependent policies when $K = 4$ (a) and $K = 3$ (b). Policies $\pi_1, \pi_2, \ldots$ (left panels) and mixture weights $w_k(s)$ (right panel) are shown for state $s = 141$ in the grid-world. **(c)** Averaged component entropy decreases as a function of $\alpha$ and for increasing $K$ (different lines), showing enhanced specialization. **(d)** Cumulative mixture entropy increases with the number of components $K$ and for higher $\alpha$ (different lines). Solid curves: averages over five random seeds; shaded regions: their variance. Dashed line in panel (d) indicates MOP, the theoretical limit.

and span the full action space. The optimal mixture policy is non-uniform: the weights associated to each component are not equal (Fig. 2a,b). This is because taking an action that steps into a terminal state collapses future entropy to zero, so cells bordering terminals autonomously assign vanishing probability to those transitions despite them remaining admissible, and this effect propagates to parent cells iteratively. The non-uniformity of the optimal mixture policy does not critically depend on having state-dependent primitive action sets, as when all cells have the same number of actions regardless of wall distances the same non-uniformity holds (Fig. 5 in Appendix). In general, policy components with higher weight tend to be broader than the ones that are chosen less frequently (see second component in Fig. 2b). Notably, the agent generates safe component policies in the sense that the probability of falling into a terminal state is close to zero, a fact that strongly shapes the final mixture policy, making it non-uniform. As expected, the entropy of the component policies declines with increased $\alpha$ and when increasing the number of components $K$ (Fig. 2c), enhancing their specialization. At the same time, the cumulative entropy of the mixture policy increases with $K$ regardless of $\alpha$ (Fig. 2d) and approaches the theoretical limit imposed by MOP (dashed line; MOP is equivalent by definition to $\alpha = 0$ and $K = 1$). When $K = 1$, the problem reduces to MOP for all $\alpha$.

## 3 COMPONENT LEARNING IN CONTINUOUS CASE

Our discrete grid-world experiments establish theoretical feasibility and validate the maxi-mix-mini-com entropy objective. Now, we extend it to the continuous-action setting. First, we formalize the continuous-action problem and present a training procedure for Gaussian mixture policies (see Fig. 1c). Next, we analyze the resulting components, quantifying the framework's ability to control action spanning and diversity. Because the continuous-action joint-entropy objective is entirely unsupervised (no extrinsic rewards are available), the standard policy-gradient theorem does not apply directly, but we can adapt it to our objective function with the Extended Policy Gradient Theorem:

**Theorem 3.** *Let $\pi_{m,\theta}(a|s)$ be a mixture policy over continuous actions parameterized by $\theta$, where $\theta$ denotes the learnable parameters of the neural networks representing the component policies $\pi_{k,\theta}$ and mixture weights $w_{k,\theta}$. Defining the state-value function as in Eq. 3, the gradient of the performance objective $J(\theta) = V(s_0)$ admits the form*

$$\nabla_\theta J(\theta) = \int_{s,a} d_{\pi_{m,\theta}}(s)\pi_{m,\theta}(a|s)\Big(\nabla_\theta \log \pi_{m,\theta}(a|s)Q(s,a) + \nabla_\theta R(s; \pi_\theta, w_\theta)\Big)\, da\, ds.$$

*where $d_{\pi_{m,\theta}}(s)$ is the discounted state occupancy measure under $\pi_{m,\theta}$, $\nabla_\theta R(s; \pi_\theta, w_\theta) = \nabla_\theta \mathcal{H}(\pi_{m,\theta}(\cdot|s)) - \alpha \nabla_\theta \sum_{k=1}^{K} w_{k,\theta}(s) \, \mathcal{H}(\pi_{k,\theta}(\cdot|s))$, $\pi_\theta = \{\pi_{1,\theta}, ..., \pi_{K,\theta}\}$ and $w_\theta = \{w_{1,\theta}, ..., w_{K,\theta}\}$.*

*Proof.* See Appendix Sec. A.3. There are several methods for optimizing the policy objective $J(\theta)$ using policy gradient RL. In our setting, the Q-function is represented by a differentiable neural network, enabling the use of the reparameterization gradient estimator for each Gaussian policy component. This estimator is unbiased, achieves lower variance than the likelihood-ratio approach by leveraging deterministic gradient computations through both the policy and Q-function networks (Haarnoja et al., 2018; Xu et al., 2019), and adapts effectively to our context.

To enable efficient gradient-based optimization, we employ the reparameterization trick (Kingma & Welling, 2013), which implements sampling from each component policy $\pi_{k,\theta}(a \mid s)$ via a deterministic mapping of an external noise variable. Concretely, actions are generated as $a = f_\theta(\varepsilon; s, k)$ with $\varepsilon \sim p(\varepsilon)$, where $f_\theta$ is a deterministic, differentiable transformation parameterized by $\theta$ and $p(\cdot)$ is a fixed noise distribution (e.g., standard Gaussian), so that gradients can be computed by backpropagation through $f_\theta$ and the Q-function. Our derivation builds on the entropy-regularized half-reparameterization policy-gradient theorem for Gaussian mixture policies introduced by He et al. (2025), which provides a low-variance estimator for the standard entropy-regularized actor–critic objective with extrinsic rewards and a soft value function.

We formalize this in the following theorem:

**Theorem 4** (Component-Reparameterization Policy Gradient Theorem). *Let $\hat{f}_\theta = f_\theta(\varepsilon; s, k)$, then the gradients of objective $\nabla_\theta J(\pi_{m,\theta})$ function under our reparametrization become*

$$
\mathop{\mathbb{E}}_{\substack{s \sim d_{\pi_{m,\theta}} \\ k \sim Cat(w_{k,\theta}(s)) \\ \varepsilon \sim p}} \left[ \nabla_\theta \log w_{k,\theta}(s) \left( Q_{\pi_{m,\theta}}(s, \hat{f}_\theta) - \log \pi_{m,\theta}(\hat{f}_\theta|s) + \alpha \log \pi_{k,\theta}(\hat{f}_\theta|s) \right) \right.
$$
$$
\left. + \nabla_\theta \hat{f}_\theta \, \nabla_a \left( Q_{\pi_{m,\theta}}(s, a) - \log \pi_{m,\theta}(a|s) + \alpha \log \pi_{k,\theta}(a|s) \right) \Big|_{a=\hat{f}_\theta} \right], \tag{16}
$$

*where $d_{\pi_{m,\theta}}(s)$ is the discounted occupancy measure under $\pi_{m,\theta}$.*

*Proof.* See Appendix Sec. A.3.

Since the mixture weights $w_\theta(\cdot)$ cannot be directly reparameterized, we adopt the straight-through Gumbel–Softmax trick from Jang et al. (2016) to produce a biased but differentiable sample. First, draw $g_1, \ldots, g_K \overset{\text{i.i.d.}}{\sim} \text{Gumbel}(0, 1)$ and form the "soft" sample

$$
y_k = \frac{\exp\big((\log w_{k,\theta}(s) + g_k)/\tau\big)}{\sum_{j=1}^{K} \exp\big((\log w_{j,\theta}(s) + g_j)/\tau\big)}, \quad k = 1, \ldots, K,
$$

where $\tau$ is the temperature parameter controlling the smoothness of the distribution. A discrete sample is obtained as

$$
\hat{z} = \text{one\_hot}\Big(\arg\max_k (\log w_{k,\theta}(s) + g_k)\Big), \quad [\text{one\_hot}(i)]_k = \delta_{k,i}.
$$

Applying the straight-through estimator yields

$$
z_k = \hat{z}_k + y_k - \text{StopGrad}(y_k), \quad \text{StopGrad}(y) = y, \ \nabla_y \text{StopGrad}(y) = 0,
$$

so that gradients flow through $y_k$ but not through its stopped version. Finally, combining $z = (z_1, ..., z_K)$ with reparameterized samples $f_\theta(\varepsilon; s, k)$ from each component gives $a = \sum_k z_k \, f_\theta(\varepsilon; s, k)$.

## 3.1 EXPERIMENTS AND RESULTS

Next, we evaluate behavioral tokenization with the maxi-mix-mini-com objective in continuous action spaces. We first used the MuJoCo Swimmer benchmark (Todorov et al., 2012), which comprises three

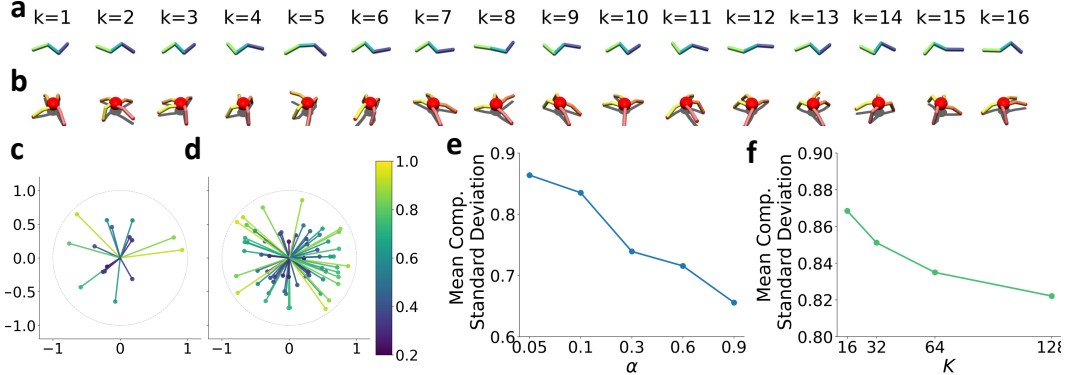

Figure 3: **(a)** Swimmer and **(b)** Ant poses from continuously applying the mode of each learned component as quantized action, showing diversity and action spanning. **(c,d)** Swimmer component action mode vectors (means $\mu_k(s)$ of the Gaussian policies) at a representative random state sampled from a trajectory induced by the mixture policy, for **(c)** $K = 16$ and **(d)** $K = 64$ components. **(e)** Ant ($K = 64$) mean component standard deviation vs. $\alpha$. **(f)** Ant mean standard deviation vs. $K$.

segments connected by two rotary joints in a linear chain, and has no terminal states. Our algorithm learns a set of components that are readily interpretable as stretches and contractions, of different sorts, of the body (Fig. 3a, for $K = 16$; see Fig. 10 in Appendix A.7 for $K = 64$; Videos 1). Their diversity and action spanning are better visualized by plotting the modes of each of the Gaussian components, using 16 components (Fig. 3c) or 64 components (Fig. 3d).

Second, we used the MuJoCo Ant benchmark (Todorov et al., 2012), a 3-dimensional quadruped with a central torso and four two-segment legs (8 actuated hinge joints). We consider terminal states as those cases where the torso touches the floor (e.g., a harmful fall). Like the swimmer, learned policies result in interpretable coordinated movements when deployed using their modes (Fig. 3b, for $K = 16$; see Fig. 11 in Appendix A.7 for $K = 64$; Videos 2). The components are diverse and span a broad range of reaching movements with one or two legs, in addition to adopting various bow poses, while keeping a balance to avoid falling down to the floor. As components are Gaussian, we use their standard deviation to characterize their specialization as a function of the hyperparameters. As expected, the standard deviation averaged over components ($K = 64$) decreases with $\alpha$ (Fig. 3e), indicating an increased degree of component specialization, and increasing $K$ reduces component standard deviation (and thus their entropy) (Fig. 3f).

**Components' usage**. We find that behavior never collapses to a single or few components, while leaving most components inactive: e.g., for $K = 16$, the per-component usage is $4\%$–$13\%$ (vs. $6.25\%$ under uniform weights), indicating no collapse. Broad component activation permits invocation and transition among tokens, enabling composition for downstream tasks (Videos 3).

## 4 HIERARCHICAL REINFORCEMENT LEARNING WITH PRE-TRAINED TOKENS

As previously demonstrated, our method generates diverse yet specialized component policies. We now use the learned components to test their effectiveness and reusability in solving different tasks by measuring cumulative reward (see Fig. 1c). Action space is quantized using the modes of the learned component Gaussians, a method that provides the first unsupervised online AQ algorithm. Conceptually, the behavioral tokens are the component policies $\pi_k(\cdot|s)$ themselves; the discrete controller selects a token index $k$, which can be grounded into a continuous action either by sampling $a_t \sim \pi_k(\cdot|s_t)$ or by executing the mode $\mu_k(s_t)$. We adopt the latter approach for stability and compatibility with standard discrete RL (see Appendix A.4 for detailed justification).

Next, state-dependent quantized actions are used as discrete actions in a DQN (Mnih et al., 2013) or discrete PPO (Schulman et al., 2017). The learned components and quantization are not allowed to change during the optimization of cumulative reward using the discrete controller (e.g. DQN); only the hierarchical policy on top of them is optimized. This deliberate freezing ensures that downstream

Table 1: Mean per-episode reward ($\pm 95\%$ CI) on Ant locomotion tasks using the same 64 fixed components.

| Method | Directional Avg. | Maze |
|---|---|---|
| SAC (continuous) | $6236.42 \pm 226.48$ | $11.99 \pm 0.51$ |
| PPO (continuous) | $3176.44 \pm 224.21$ | $3.64 \pm 0.57$ |
| Ours ($K = 64$ discrete) | $3312.54 \pm 235.23$ | $9.55 \pm 0.71$ |

Table 2: Performance ($\pm 95\%$ CI) of our and unsupervised skill discovery algorithms on Ant & Humanoid with $K = 64$.

| Method | Ant (K=64) | Humanoid (K=64) |
|---|---|---|
| Ours | $3498.27 \pm 186.08$ | $5672.19 \pm 62.49$ |
| DADS | $124.85 \pm 16.64$ | $2516.38 \pm 67.30$ |
| LSD | $1320.54 \pm 87.12$ | $1088.69 \pm 410.57$ |
| DIAYN | $341.84 \pm 26.02$ | $661.70 \pm 97.43$ |
| METRA | $443.46 \pm 121.00$ | $1024.60 \pm 49.57$ |

Table 3: Mean ($\pm 95\%$ CI) returns for SAC and our method (K=16, K=64) across continuous-control tasks. Our method uses a discrete PPO high-level controller for all environments except FetchReach-v2, where DQN is used.

| Environment | SAC | PPO | Ours $K = 16$ | Ours $K = 64$ |
|---|---|---|---|---|
| **Locomotion (MuJoCo)** | | | | |
| Humanoid-v5 | $6628.26 \pm 232.03$ | $2285.35 \pm 948.65$ | $5360.89 \pm 108.05$ | $5672.19 \pm 62.49$ |
| Ant-v5 | $6236.42 \pm 226.48$ | $3176.44 \pm 224.21$ | $2515.53 \pm 308.08$ | $3498.27 \pm 186.08$ |
| Walker2d-v5 | $4790.21 \pm 130.33$ | $3834.31 \pm 160.21$ | $2394.14 \pm 63.61$ | $2473.93 \pm 234.04$ |
| Hopper-v5 | $3516.70 \pm 116.79$ | $2825.80 \pm 176.51$ | $3010.87 \pm 145.66$ | $3226.50 \pm 119.30$ |
| Swimmer-v5 | $106.49 \pm 4.43$ | $98.07 \pm 6.58$ | $109.96 \pm 3.80$ | $119.05 \pm 3.03$ |
| **Other Continuous Control** | | | | |
| BipedalWalker-v3 | $307.74 \pm 5.50$ | $281.27 \pm 8.57$ | $265.69 \pm 15.37$ | $284.93 \pm 11.96$ |
| LunarLanderContinuous-v3 | $256.28 \pm 2.11$ | $217.68 \pm 22.45$ | $128.81 \pm 35.22$ | $220.38 \pm 13.06$ |
| FetchReach-v2 | $0.83 \pm 0.15$ | $0.80 \pm 0.18$ | $1.00 \pm 0.00$ | $1.00 \pm 0.00$ |

returns strictly measure how well the unsupervised behavioral bank captures the dynamics, preventing fine-tuning on task rewards from obscuring the contribution of the shared behavioral prior.

**Transfer of learned behavioral tokens across tasks**. We first used two very different locomotion tasks on the MuJoCo Ant: maximizing torso velocity along one of the 4 cardinal directions (directional locomotion task, with four conditions), and reaching a central goal in a maze from a random perimeter start (maze navigation task). These two tasks a priori need different skills, such as running (in the directional locomotion task) and steering the torso and turning in different directions (in the maze navigation tasks). We compared the performance of our algorithm against SAC (Haarnoja et al., 2018) and standard continuous PPO (Schulman et al., 2017) (Table 1; see Appendix A.6 for training details). With just $K = 64$ components, our method recovers 56% of SAC's performance and outperforms continuous PPO by nearly an order of magnitude on the directional locomotion tasks. Furthermore, using these same components, its performance remains comparable to the continuous baselines in the maze navigation task, indicating that the learned components generalize effectively across different locomotion challenges. This result is remarkable: the components have not been specialized or fine tuned to either of the two tasks. Thus, the learned components are reusable, diverse enough to solve different tasks, and flexible to be composed to generate complex sequential behavior.

**Comparison with continuous control**. We evaluate the learned components on standard continuous control benchmarks comprising locomotion tasks with high- and low-dimensional action spaces, together with one robotic manipulation environment. We find that in Fetch Reach and Swimmer the maxi-mix-mini-com entropy objective not only matches but surpasses state-of-the-art continuous methods (Table 3). This is because the discovery of just a few quantized actions that span action space while avoiding bottlenecks induced by the non-linearity of their dynamics facilitates the search and selection of the correct actions by a hierarchical policy acting on top of them. For higher dimensional action spaces, such as Humanoid with 17 degrees of freedom, we find 79% performance with just $K = 16$ components. These results show that our method scales up to high-dimensional action control, hard-to-balance problems without increasing the number of components.

**Comparison with random action quantization**. It is important to test whether our learned components are better than randomized AQ baselines. We chose (a) uniform sampling across the continuous action space; (b) sampling from a zero-mean Gaussian distribution with standard deviation $0.2$ and (c) per-dimension i.i.d. sampling from the discrete alphabet $\{-m, 0, m\}$. Our method achieves higher performance across all evaluated cases (Tables 5,6 in Appendix A.7). These baselines span the natural hand-designed quantizers (Gaussian, uniform, and a per-dimension uniform three-bin alphabet), and Tables 5–6 show that they consistently lag behind our learned components, reinforcing that principled behavioral tokenization—rather than naïve random quantization—is necessary to span

action space effectively. For instance, in the high-dimensional Humanoid and Walker agents, random AQ approaches entirely fail to prevent the agent from entering terminal states (e.g., falling; Videos 4).

**Comparison with skill-discovery methods**. We compared our method to four leading unsupervised skill-discovery (USD) methods. For our approach, we used a mixture with $K = 64$; for each USD baseline we adopted a 64-dimensional discrete latent space. In both cases the state representation omitted the agent's $x, y$ positional coordinates, and forward-locomotion performance was evaluated using discrete PPO. We find that our method attains higher performance than the USD baselines (Table 2). We attribute this difference to objective design: USD methods are primarily intended to discover temporally extended behaviors that emphasize partitioning the state space into distinct skills, whereas our method explicitly promotes per-state diversity and specialization of action components. These quantitative findings are corroborated by qualitative behavioral comparisons between the behavior generated by each component of our mixture policy and the behavior of the latent-conditioned skill policy when using a fixed latent skill $z$ (Videos 5, Appendix A.7): our components exhibit greater diversity and a broader repertoire of behavioral tokens required to solve downstream tasks.

**Effect of specialization coefficient $\alpha$ and number of components $K$**. As said above, our method does not require a large number of components to recover or exceed the performance of a continuous controller. Increasing $K$ does not produce proportional gains, e.g., raising $K$ from 16 to 128 increases performance by only 22% (Table 6). This result shows that the agent needs only few components to effectively span action space. The specialization coefficient $\alpha$ has limited impact on performance once $K$ is sufficient to solve the task ($K \geq 64$), with effects becoming particularly marginal at $K = 128$. Thus, a moderate component count ($K \approx 64$) combined with mid-low values of $\alpha$ (e.g., 0.2) is sufficient to obtain near-peak performance while avoiding unnecessary computational overhead. Finally, increasing $K$ does not affect pre-training time, so learning more components is never a bottleneck: empirically, wall-clock runtimes for $K = 16, 64$ and $128$ were the same for Ant-v5 (all between 3.4 and 3.6 Ksec for 200K interactions, averaged over 5 seeds for each $K$).

## 5 DISCUSSION AND CONCLUSION

Our paper has tackled an important, yet overlooked, problem: how to generate a bank of behavioral tokens that are task-agnostic, state-dependent, action-centric and learnable in an unsupervised manner. We have presented an online unsupervised method for behavioral tokenization using the maxi-mix-mini-com entropy objective. This objective generates diverse behavioral tokens because the mixture policy aims to maximize cumulative action entropy. The objective also promotes the discovery of distinct tokens because of the per-component entropy penalty. We have provided the first unsupervised online AQ algorithm, which takes the modes of the learned components as quantized actions. Our AQ algorithm, when combined with discrete-action controllers, provided comparable results to state-of-the-art continuous controllers and outperformed random quantization and skill-discovery methods. Importantly, our algorithm does not trivially collapse to uniform policies assigning the same probability to every action: although the objective is maximizing joint entropy, Eq. 5, the presence of (i) terminal states and/or (ii) non-linear dynamics makes the policy to steer away from uniformity. Finally, an exciting avenue for future work is to relax the frozen-component assumption and explore joint fine-tuning on downstream rewards. Because Eq. 2 is compatible with extrinsic returns (Sec. 2), one could combine maxi-mix-mini-com with task rewards to fine tune the components themselves, treating the intrinsic term as a regularizer that preserves diversity while improving task-specific competence (Peters et al., 2010).

**Limitations**: Our online learning method faces practical challenges. Notably, we often need to downscale the variance of the learned component policies to make them compatible with discrete actions. We also resort to taking the modes of these components for AQ. However, this is not an inherent flaw in our algorithm—by increasing the number of components $K$, the standard deviation naturally decreases, rendering the policies more suitable for discrete-policy reward maximization. Downscaling is a technique also utilized in concurrent work, such as (Sheebaelhamd et al., 2025), to mitigate similar challenges. In addition, unlike recent skill discovery methods such as METRA (Park et al., 2024), our approach does not currently support zero-shot downstream task solving and relies on a discrete controller to compose behavioral tokens into meaningful behaviors.

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

# A  APPENDIX

## A.1  RELATED WORK

**Quantization of Continuous Action Spaces**: In control theory, the idea of discretizing (or quantizing) continuous action spaces dates back to Bushaw's work on discontinuous forcing terms (Bushaw, 1953), and was formalized in the "bang–bang" control problem by Bellman et al. (1956). However, naive uniform discretization suffers from an exponential explosion of possible actions in high-dimensional settings. To mitigate this, a family of methods assumes independence across action dimensions, discretizing each separately (Tavakoli et al., 2018; Andrychowicz et al., 2020; Vieillard et al., 2021; Tang & Agrawal, 2020). Similarly, Seyde et al. (2021) replace Gaussian policies with factorized Bernoulli "bang–bang" policies and show that such extreme per-dimension discretization can still solve standard continuous-control benchmarks. Complementary approaches instead model inter-dimensional dependencies—either via sequential prediction of continuous actions (Metz et al., 2017), distributional policy optimization frameworks (Tessler et al., 2019), learned Q-network structures for high-dimensional control (Sakryukin et al., 2020), or hypergraph representations of action-value functions (Tavakoli et al., 2020).

**Offline and Demonstration-Based Action Quantization**: Concurrently, offline and demonstration-based approaches have focused on deriving discrete action primitives directly from fixed data (not allowing the interaction of the agent with the environment). For example, state-conditioned action quantization has been implemented via VQ-VAE frameworks (Luo et al., 2023). Dadashi et al. (2021) proposed AQuaDem, which constructs a state-conditioned discretization of the continuous action space entirely offline using logged human demonstrations before any online reinforcement learning. In offline settings, an action-quantization objective partitions a finite set of actions drawn directly from the dataset; it does not discover or generate novel behaviors. Concretely, for each state $s$, the algorithm selects a small "codebook" of prototype actions from the empirical action support. As a result, its ability to propose useful or task-optimized actions is strictly bounded by the diversity and quality of the recorded trajectories. This limitation is illustrated in Luo et al. (2023) (their Table 5), where walker2d-medium-replay-v2 — and similarly halfcheetah and hopper — exhibit very poor performance when the offline dataset lacks expert coverage, while performance improves when expert demonstrations are included (walker2d-expert-v2). Therefore, unless high-quality actions are already present in the dataset or the dataset is extremely large, offline quantization alone has limited capacity to produce task-useful action sets.

In contrast, in the online setting considered in our current paper, a quantization method is not confined to a pre-collected action set. The objective shifts from "compressing" a static action distribution to "constructing" a repertoire of actions that are both diverse and high-reward for downstream tasks. Online quantization can actively propose and refine actions through interaction, thereby learning useful actions rather than merely re-quantizing observed behaviors. Q-Transformer (Chebotar et al., 2023) adopts a per-dimension discretization that treats each action axis as a separate time step, predicting one bin at a time. This avoids the exponential growth of actions in multi-dimensional discretization, but it forces a coarse–fine trade-off in movement granularity. This important limitation has motivated contemporaneous work on quantization-free continuous sequence models such as the Q-FAT, which directly parameterize continuous actions via Gaussian mixture models, eliminating per-dimension binning (Sheebaelhamd et al., 2025). However, like Q-Transformer's per-dimension discretization scheme, Q-FAT is trained entirely on offline demonstration datasets using a negative log-likelihood objective without access to reward signals, making it susceptible to distributional shift and dataset bias when deployed out-of-distribution (Chang et al., 2021). Moreover, there is no guarantee that the mixture-model representation will maintain diverse component utilization (see e.g., (Sheebaelhamd et al., 2025)). To fill in these gaps, in this work we have introduced the first unsupervised online AQ method, which does not require offline data nor any task definition.

**Mixture Policy Representations and Entropy Estimation**: Kolchinsky & Tracey (2017) introduce a family of analytic estimators for the differential entropy of mixture distributions, based on pairwise distance functions between components. He et al. (2025) examines conditional Gaussian mixture policies within the entropy-regularized actor-critic framework, derives both half-reparameterization and Gumbel-softmax gradient estimators for the mixture weights and components, and demonstrates that mixture policies yield in some cases more robust exploration and higher solution quality than unimodal Gaussians across a variety of benchmarks. Daniel et al. (2016) extend the Relative Entropy

Policy Search algorithm to hierarchical "mixed-option" policies by formulating policy learning as a latent-variable problem and incorporating an uncertainty constraint to enforce mode separation. Learned components are specific to a task, and therefore they do not generalize to new tasks or environments. Nematollahi et al. (2022) propose a hybrid approach in which a Gaussian mixture dynamical system is first learned from demonstrations to model trajectory distributions of robot skills. Importantly, the half-reparameterization estimator of He et al. (2025) targets the standard entropy-regularized RL objective where policy improvement is driven by extrinsic rewards and a soft value function. Our maxi-mix–mini-com objective, in contrast, is entirely unsupervised: it maximizes mixture entropy while concurrently penalizing component entropies so that the learned tokens remain diverse yet specialized. Theorem 4 adapts the half-reparameterization construction to this reward-free objective, ensuring that mixture-policy gradients remain tractable even when the only learning signal is the opposing pair of intrinsic entropy terms. This distinction guarantees that the resulting components are dynamics- and embodiment-specific primitives agnostic to any downstream reward.

**Hierarchical RL** encompasses frameworks such as goal-conditioned RL, unsupervised skill discovery, and option-based methods that decompose tasks into high- and low-level policies (Sutton & Barto, 1998). Goal-conditioned approaches like Universal Value Function Approximators train policies to achieve explicit goals but require predefined goal signals Schaul et al. (2015), while skill-discovery methods (e.g., DIAYN (Eysenbach et al., 2019), VALOR (Achiam et al., 2018)) uncover diverse behaviors via mutual-information or empowerment objectives but often lack convergence guarantees or principled composition. Option-based frameworks learn temporally extended primitives under formal hierarchies but can depend on offline data or manual subgoal design (Bacon et al., 2017). In contrast, our maxi-mix-mini-com framework discovers entropy-driven component policies fully online, providing provable convergence, explicit diversity control, and interpretable components that integrate seamlessly into hierarchical architectures.

**Intrinsic Motivation** addresses the question of what reward-agnostic signals can generate behavior useful for learning and discovery. Several frameworks have emerged where intrinsic signals are major drives of behavior: information seeking (Gottlieb et al., 2013), novelty seeking (Lehman & Stanley, 2011), empowerment (Klyubin et al., 2005; Leibfried et al., 2019), minimizing free energy (Friston et al., 2013), or occupying action-state space (Ramírez-Ruiz et al., 2024). We have focused on the latter because the maximum occupancy principle (MOP) provides a principled way to generate diverse, complex behaviors limited only by the constraints of the agent and the presence of terminal states. Empowerment maximizes the mutual entropy between future states and sequences of actions, and therefore it is an appealing objective to generate diverse behavior. However, it has been shown that empowerment provides no clear closed-form recursion beyond its per-step formulation (Moreno-Bote & Ramirez-Ruiz, 2023). In any event, it is worth exploring in future work different intrinsic motivation objectives for action-policy discretization.

**Unsupervised skill discovery (USD)** methods commonly learn a latent-conditioned policy $\pi(a|s,z)$ with a skill latent $z$ sampled once per episode, and optimize intrinsic objectives that couple $z$ to states, short transitions, terminal states, or entire trajectories rather than to per-step action choices. For example, DIAYN maximizes $I(S;Z)$ to induce separable state marginals for each skill, VIC maximizes episode-scale dependence $I(S_T;Z \mid S_0)$ to enlarge the set of reliably reachable terminations, and VALOR decodes skill identity from full trajectories (Eysenbach et al., 2019; Gregor et al., 2017; Achiam et al., 2018). Other methods explicitly target short-horizon dynamics: DADS maximizes $I(S';Z \mid S)$ via a skill-conditioned dynamics model, and CIC formulates a contrastive objective on single-step transitions while regularizing transition entropy to preserve coverage (Sharma et al., 2020; Laskin et al., 2022). Recent work has incorporated geometric or metric structure into the dependency measure (e.g., enforcing Lipschitz/non-expansive embeddings or using Wasserstein dependency measures) to bias learned skills toward dynamic, far-reaching behaviors (Park et al., 2022; 2024). Because these objectives encourage distinguishability at the level of states, transitions, terminations, or trajectories, distinct skills can select identical atomic actions in the same state and remain differentiable only by their subsequent evolution; consequently, USD methods are by design temporally-extended primitives sampled per episode and do not enforce per-timestep action-level diversity, which limits their suitability for tasks that require tightly interleaved, within-state switching between atomic actions.

In summary, a recurring limitation of prevailing USD objectives is that they couple the latent skill primarily to *where* an agent goes (state marginals or terminal states) rather than to *how* it behaves at

shared states (characteristic action patterns, magnitudes, or variability). Therefore, USD objectives are *state-centric*, while our maxi-mix-mini-com objective is *action-centric*. Operationally, for example, *running* toward two different goal states may instantiate the same skill with differing directions, whereas *walking* and *running* along a common heading should be treated as different skills because their one-step effects (increment magnitudes and variability) differ systematically despite substantial state overlap. Objectives that maximize dependence between skills and states can therefore separate by geography or heading without ensuring that skills remain distinguishable when trajectories overlap. For reuse and composition, it is therefore desirable to require identifiability from local behavior under shared state distributions rather than solely from global or terminal state statistics. Our maxi-mix-mini-com entropy-based algorithm accomplishes this goal: it leads to the discovery of action-spanning, state-dependent, single-step, distinguishable and composable behavioral tokens.

## A.2 MONOTONIC POLICY ITERATION FOR THE MAXI-MIX-MINI-COMP ENTROPY OBJECTIVE

Here we provide a full proof of Theorems 1 and 2. We reproduce some results previously shown in Sec. 2 to ease understanding. First we remind the reader that our objective can be expressed as maximizing the value function over component and coefficients as the optimality Bellman equation

$$V^*(s) = \max_{\pi,w} \left( R(s;\pi,w) + \gamma \sum_a \pi_m(a|s)Q^*(s,a) \right) \tag{17}$$

$$R(s;\pi,w) \equiv \mathcal{H}(\pi_m(\cdot|s)) - \alpha \sum_k w_k(s)\mathcal{H}(\pi_k(\cdot|s)) , \tag{18}$$

where $\pi$ indicates the set of component policies $\pi = \{\pi_1, ..., \pi_K\}$ and $w = \{w_1, ..., w_K\}$, where the conditioning on the state $s$ is omitted but understood. Note that the sum limits in $\sum_a$ are state-dependent, as we in general allow for state-dependent action set $\mathcal{A}(s)$.

If $\alpha < 1$, the immediate reward is non-negative, $R(s;\pi,w) \geq 0$, because the entropy of a mixture is always larger or equal than its maximum entropy component – easy to prove using concavity of $-x\log x$ in $[0, 1]$. Since the immediate reward $R(s;\pi,w)$ is bounded for all $s$ in finite action spaces, it is not difficult to show that the optimal value function $V^*$ admits an unique solution, although possibly through non-unique optimal $\pi$ and $w$ (see e.g., steps in (Haarnoja et al., 2018), using the fact that if $0 \leq \gamma < 1$, then the optimal Bellman operator is a contraction).

We first prove that for a high capacity mixture policy, that is, when the number of components is larger or equal than the action space, then solving Eq. 17 is equivalent to solving an unconstrained problem with no restrictions on the mixture policy.

**Theorem 1.** *If $K(s) \geq |\mathcal{A}(s)|$ for all $s$, then the optimal solution of Eq. 17 is such that $\pi_k(a_k|s) = 1$ only for one action $a_k$ and the set of actions spans $\mathcal{A}(s)$, that is, $\{a_k\} = \mathcal{A}(s)$. Then, $\pi_m$ in Eq. 1 is unconstrained and the optimal value function obeys*

$$V^*(s) = \max_{\pi,w} \left( \mathcal{H}(\pi_m(\cdot|s_t)) + \gamma \sum_a \pi_m(a|s)Q^*(s,a) \right) \tag{19}$$

*The optimal mixture policy is unique, while the mixture components are not unique if $|\mathcal{A}(s)| > 1$ at least for one state.*

*Proof.* It is obvious that for any fixed mixture policy $\pi_m$, the value function in Eq. 3 is maximized if each of the components can be made to have zero entropy (as $\alpha > 0$), eliminating the negative contribution to the immediate reward. Zero entropy components can be obtained iff for each component $k$, there exists a single action $a_k$ such that $\pi_k(a_k|s) = 1$. As $K(s) \geq |\mathcal{A}(s)|$, and defining $\mathcal{A}(s) = \{a_1, ..., a_{\mathcal{A}(s)}\}$, we can choose the components $\pi_k$ such that $\pi_k(a_k|s) = 1$ (that is, it is a deterministic component policy) for $k = 1, ..., |\mathcal{A}|(s)$, and $\pi_k(a_1|s) = 1$ for $k > |\mathcal{A}|(s)$. Then, any policy $\pi(a|s)$ can be built from these components using $w_k(s) = \pi(a_k|s)$ in Eq. 1 for $k \leq |\mathcal{A}|(s)$ and $w_k(s) = 0$ otherwise. This means that the mixture policy is unconstrained, and then it can be made to optimize the Bellman Eq. 5 with $\alpha = 0$, or equivalently Eq. 19. Non uniqueness of the components is clear because we are free to permute assignments between actions and components. Uniqueness of $V^*$ and the optimal policy $\pi_m^*$ has been shown in Ramírez-Ruiz et al. (2024). $\square$

For the case $K(s) < |\mathcal{A}(s)|$, we can write the optimality Bellman equation 17 (see Sec. 2) as

$$V^*(s) = \max_{\pi,w} \max_r \left( G(s; \pi, w, r) + \gamma \sum_a \pi_m(a|s) Q^*(s,a) \right) \qquad (20)$$

Our algorithm (see Sec. 2) proceeds by following the next steps:

(1) Initial conditions: Start the value function $V^{(0)}(s) = 0$ for all $s$ and arbitrary $\pi^{(0)} = \{\pi_1^{(0)}, ..., \pi_K^{(0)}\}$ and $w^{(0)} = \{w_1^{(0)}, ..., w_K^{(0)}\}$ outside the simplex boundaries, that is, $\pi_k^{(0)}(a|s) > 0$ and $w_k^{(0)}(s) > 0$ for all $k$, $s$ and $a$. Define the initial responsibilities as

$$r_k^{(0)}(s,a) = \frac{w_k^{(0)}(s)\, \pi_k^{(0)}(a|s)}{\pi_m^{(0)}(a|s)} > 0 \,. \qquad (21)$$

Iterate the following step (2) for $n = 1, 2, ....$

(2) At iteration $n$, start from $V^{(n-1)}$, $\pi^{(n-1)}$, $w^{(n-1)}$ and $r^{(n-1)}$. Perform the following steps, (2a) to (2d).

(2a) Define the updated value function $V^{(n)}$ as

$$V^{(n)}(s) = G(s; \pi^{(n-1)}, w^{(n-1)}, r^{(n-1)}) + \gamma \sum_a \pi_m^{(n-1)}(a|s) Q^{(n-1)}(s,a) \,, \qquad (22)$$

with $Q^{(n-1)}(s,a) = \sum_{s'} p(s'|s,a) V^{(n-1)}(s')$ and $\pi_m^{(n-1)} = \sum_k w_k^{(n-1)}(s) \pi_k^{(n-1)}(a|s)$.

Note that with this definition, we have $V^{(1)}(s) \geq V^{(0)}(s) = 0$ for all $s$, because the gain $G(s; \pi^{(0)}, w^{(0)}, r^{(0)}) = R(s; \pi^{(0)}, w^{(0)})$ is non-negative.

(2b) Compute $\pi^{(n)}$ fixing $V^{(n-1)}$ and $r^{(n-1)}$ using

$$\pi_k^{(n)}(a|s) = \frac{[r_k^{(n-1)}(s,a)]^{\frac{1}{1-\alpha}} e^{\frac{\gamma}{1-\alpha} Q^{(n-1)}(s,a)}}{\sum_{a'} [r_k^{(n-1)}(s,a')]^{\frac{1}{1-\alpha}} e^{\frac{\gamma}{1-\alpha} Q^{(n-1)}(s,a')}} \,. \qquad (23)$$

This equation is the result of the optimization

$$\pi^{(n)}(\cdot|s) = \arg\max_\pi \left( G(s; \pi, w^{(n-1)}, r^{(n-1)}) + \gamma \sum_{k,a} w_k^{(n-1)}(s)\, \pi_k(a|s) Q^{(n-1)}(s,a) \right) \,, \quad (24)$$

where $\pi^{(n)}(\cdot|s) = \{\pi_1^{(n)}(\cdot|s), \cdots, \pi_K^{(n)}(\cdot|s)\}$, as it can be checked adding Lagrange multipliers and taking derivatives (see details in Appendix A.2.1). Note that the ordering of the component policies $\pi_1, \ldots, \pi_K$ is immaterial, as the objective in Eq. 24 is additive over $k$, and each $\pi_k$ can be optimized independently when $w^{(n-1)}$ and $r^{(n-1)}$ are fixed. Consequently, the solution is invariant to any permutation of the indices. The solution $\pi_k^{(n)}(a|s)$ provided in Eq. 23 is the unique global optima if $w_k^{(n-1)}(s) > 0$ for all $k$ and $s$, as then the solution lies inside the simplex and the objective is strictly concave.

Defining $V_\pi^{(n)}$ as the argument of the right hand side of Eq. 24, after replacing $\pi$ with the optimal $\pi_k^{(n)}(a|s)$, we obtain

$$V_\pi^{(n)} = G(s; \pi^{(n)}, w^{(n-1)}, r^{(n-1)}) + \gamma \sum_{k,a} w_k^{(n-1)}(s)\, \pi_k^{(n)}(a|s) Q^{(n-1)}(s,a) \qquad (25)$$

$$= (1-\alpha) \sum_k w_k^{(n-1)}(s) \log \left[ \sum_a [r_k^{(n-1)}(s,a)]^{\frac{1}{1-\alpha}} e^{\frac{\gamma}{1-\alpha} Q^{(n-1)}(s,a)} \right]$$

$$- \sum_k w_k^{(n-1)}(s) \log(w_k^{(n-1)}(s)) \,. \qquad (26)$$

Naturally, we have $V_\pi^{(n)}(s) \geq V^{(n)}(s)$ for all $s$, because after the optimization the quantity in the bracket on right hand side of Eq. 24 cannot decrease.

(2c) Compute $w^{(n)}$ fixing $V^{(n-1)}$ and $r^{(n-1)}$ using

$$w_k^{(n)}(s) = \frac{\left(\sum_a [r_k^{(n-1)}(s,a)]^{\frac{1}{1-\alpha}} e^{\frac{\gamma}{1-\alpha}Q^{(n-1)}(s,a)}\right)^{1-\alpha}}{\sum_l \left(\sum_a [r_l^{(n-1)}(s,a)]^{\frac{1}{1-\alpha}} e^{\frac{\gamma}{1-\alpha}Q^{(n-1)}(s,a)}\right)^{1-\alpha}} \cdot \tag{27}$$

This equation is the result of the optimization

$$w^{(n)}(s) = \arg\max_w \left( G(s; \pi^{(n)}, w, r^{(n-1)}) + \gamma \sum_{k,a} w_k(s) \pi_k^{(n)}(a|s) Q^{(n-1)}(s,a) \right), \tag{28}$$

by using the previously computed $\pi^{(n)}$ in step (2b), and where we denote $w^{(n)}(s) = \{w_1^{(n)}(s), \cdots, w_K^{(n)}(s)\}$. The solution in Eq. 27 is the only global maximum provided that $r_k^{(n-1)}(s,a) > 0$ for all $k$, $s$ and $a$ (see Appendix A.2.3 for details).

Defining $V_{\pi,w}^{(n)}$ as the argument of the right hand side of Eq. 28, after replacing $w$ with the optimal $w_k^{(n)}(s)$, we obtain

$$V_{\pi,w}^{(n)} = G(s; \pi^{(n)}, w^{(n)}, r^{(n-1)}) + \gamma \sum_{k,a} w_k^{(n)}(s) \pi_k^{(n)}(a|s) Q^{(n-1)}(s,a) \tag{29}$$

$$= \log\left[\sum_k \left(\sum_a [r_k^{(n-1)}(s,a)]^{\frac{1}{1-\alpha}} e^{\frac{\gamma}{1-\alpha}Q^{(n-1)}(s,a)}\right)^{1-\alpha}\right]. \tag{30}$$

Naturally, we have $V_{\pi,w}^{(n)}(s) \geq V_\pi^{(n)}(s)$ for all $s$, because after the optimization the right hand side of Eq. 28 cannot decrease. This, in turn, implies that $V_{\pi,w}^{(n)}(s) \geq V^{(n)}(s)$ using the last inequality in step (2a).

(2d) Compute $r^{(n)}$ fixing $V^{(n-1)}$, $\pi^{(n)}$ and $w^{(n)}$ using

$$r_k^{(n)}(s,a) = \frac{w_k^{(n)}(s) \pi_k^{(n)}(a|s)}{\pi_m^{(n)}(a|s)}, \tag{31}$$

with $\pi_m^{(n)}(a|s) = \sum_k w_k^{(n)}(s) \pi_k^{(n)}(a|s)$. This equation is the result of the optimization (see Appendix A.2.3 for details),

$$r^{(n)}(s, \cdot) = \arg\max_r \left( G(s; \pi^{(n)}, w^{(n)}, r) + \gamma \sum_{k,a} w_k^{(n)}(s) \pi_k^{(n)}(a|s) Q^{(n-1)}(s,a) \right), \tag{32}$$

where $r^{(n)}(s, \cdot) = \{r_1^{(n)}(s, \cdot), \cdots, r_K^{(n)}(s, \cdot)\}$.

Defining $V_{\pi,w,r}^{(n)}$ as the quantity within the bracket on the right hand side of Eq. 32 after replacing $r$ with the optimal $r_k^{(n)}(s,a)$, we obtain

$$V_{\pi,w,r}^{(n)} = G(s; \pi^{(n)}, w^{(n)}, r^{(n)}) + \gamma \sum_{k,a} w_k^{(n)}(s) \pi_k^{(n)}(a|s) Q^{(n-1)}(s,a). \tag{33}$$

Naturally, we have $V_{\pi,w,r}^{(n)}(s) \geq V_{\pi,w}^{(n)}(s)$ for all $s$, because after the optimization the right hand side of Eq. 32 cannot decrease. This, in turn, implies that $V_{\pi,w,r}^{(n)}(s) \geq V^{(n)}(s)$ using steps (2a) and (2b).

**Theorem 2.** *The algorithm following steps (1) and (2), which consists in iterating the $\pi$, $w$ and $r$ using Eqs. 22,23,27,31 in this order, improves the value function monotonically until convergence.*

*Proof.* From (2a) above we have seen that $V^{(1)} \geq V^{(0)}$. Assume that it is true $V^{(n)} \geq V^{(n-1)}$ for some $n > 0$. Then, from the definition of $V^{n+1}$ using Eq. 22 we see that

$$V^{(n+1)}(s) = G(s; \pi^{(n)}, w^{(n)}, r^{(n)}) + \gamma \sum_a \pi_m^{(n)}(a|s)Q^{(n)}(s,a) \tag{34}$$

$$\geq G(s; \pi^{(n)}, w^{(n)}, r^{(n)}) + \gamma \sum_a \pi_m^{(n)}(a|s)Q^{(n-1)}(s,a) \tag{35}$$

$$= V_{\pi,w,r}^{(n)} \tag{36}$$

$$\geq V^{(n)}(s) , \tag{37}$$

where in the second line we have used the induction assumption, implying that $Q^{(n)}(s,a) \geq Q^{(n-1)}(s,a)$ for all $(s,a)$, in the third line we have used the definition of $V_{\pi,w,r}^{(n)}$ in Eq. 33, and in the last line we have used the final implication in (2d).

Then, by induction, the series $V^{(n)}(s)$ is non-decreasing for all $s$, and since the value function is finite ($V(s) \leq \log(\max_s |\mathcal{A}(s)|)/(1-\gamma)$ for any policy $\pi_m$), the series converges to a finite value $V^{(\infty)}(s)$ as $n$ goes to infinity. There is strict improvement ($>$ instead of $\geq$ in all equations above) whenever the updated $\pi$, $w$ or $r$ are different from the previous ones.

We finally check for consistency of Eqs. 23,27,31, as these optima are obtained by taking derivatives with Lagrange multipliers and assuming that the optima do not lie on the simplex boundaries. The initial condition indeed obeys this, $\pi_k^{(0)}(a|s) > 0$, $w_k^{(0)}(s) > 0$ and $r_k^{(0)}(s,a) > 0$ for all $k$, $s$ and $a$. Then, positivity of all variables is true for $n = 0$. Assuming that positivity of all variables is true for some $n$ such that $n > 0$, then it can be shown that it is again true for $n+1$ using Eqs. 23,27,31. By induction, it follows that the optima are always in the simplex but outside the boundary, although they can get infinitely close to the simplex boundary as $n$ goes to infinity, meaning that the converged solution might actually be at the boundary. $\square$

*Remark*: The limit $V^{(\infty)}(s)$ must be a solution to the Bellman Eq. 3. Therefore, it corresponds to a proper value function with a proper policy. Indeed, the policy must converge as well to a policy $\pi_m^{(\infty)}(a|s)$, because it cannot change when the value function $V^{(n)}(s)$ has converged: if the policy changed, then there should a strict ($>$) improvement of the value function in every new iteration, which contradicts the assumption that the value function has converged. However, convergence of the series $V^{(n)}(s)$ does not imply that the optimal Bellman equation is solved by $V^{(\infty)}(s)$, so it might lead to a suboptimal value and policy. The converged values also might depend on the initial conditions.

*Remark*: Any orderings of the updates for $\pi$, $w$ or $r$ lead to equivalent algorithms. This is because $\pi$ and $w$ only depend on $r$ and $Q$, so their order is interchangable, and any other update ordering is just a cyclic permutation of one of these two.

*Remark*: Our algorithm never leads to trivial solutions. In both trivial-collapse scenarios—the case where all component policies are identical and the case where the mixture weights are deterministic—the mixture distribution $\pi_m$ coincides with a single component $k$, so $\mathcal{H}(\pi_m(\cdot|s)) = \mathcal{H}(\pi_k(\cdot|s))$ and $\sum_r w_r \mathcal{H}(\pi_r(\cdot|s)) = \mathcal{H}(\pi_k(\cdot|s))$. Substituting into Eq. 18 yields the same degenerate reward

$$R(s; \pi, w) = \mathcal{H}(\pi_k(\cdot|s)) - \alpha\mathcal{H}(\pi_k(\cdot|s)) = (1-\alpha)\,\mathcal{H}(\pi_k(\cdot|s)),$$

which is strictly lower than the reward under a truly diverse mixture (high $\mathcal{H}(\pi_m)$, low $\sum_k w_k \mathcal{H}(\pi_k)$), so these solutions are never preferred. Indeed, if our algorithm is started with $w_k > 0$ for all components and with non-identical components (as in our experiments), the initial magnitude of the value function will be higher than that of the trivial solution; therefore the solution after convergence will not fall into the trivial solution.

### A.2.1 FINDING OPTIMAL COMPONENT POLICIES AT ITERATION $n$

At iteration $n$, with fixed value function $V^{(n-1)}$, mixture weights $w^{(n-1)}$ and responsibilities $r^{(n-1)}$, the update for the component policies $\pi_k$ is given by the maximization problem

$$
\pi^{(n)}(\cdot|s) = \arg\max_{\pi} \left( G(s; \pi, w^{(n-1)}, r^{(n-1)}) + \gamma \sum_{k,a} w_k^{(n-1)}(s)\, \pi_k(a|s) Q^{(n-1)}(s,a) \right).
$$

In this optimization, for a given $k$, only the following terms depending on $\pi_k$ contribute,

$$
-(1-\alpha) \sum_a \pi_k(a|s) \log \pi_k(a|s) + \sum_a \pi_k(a|s) \left[ \log r_k^{(n-1)}(s,a) + \gamma Q^{(n-1)}(s,a) \right],
$$

subject to $\sum_a \pi_k(a \mid s) = 1$. This expression is multiplied by $w_k^{(n-1)}(s)$, but because $w_k^{(n-1)}(s) > 0$ for all $k$, $s$ and $n$ (see Theorem 2), it can be removed. We note that for $\alpha < 1$ this expression is strictly concave in $\pi_k(\cdot|s)$; therefore if there is a critical point in the simplex $\sum_a \pi_k(a \mid s) = 1$, it will be the global maximum, which will be the case as shown below.

We note that the simplification that arises from having a constant weight $w_k^{(n-1)}(s)$ multiplying the previous equation does not hold if the weighted sum of entropies in Eq. 2 is replaced by an unweighted sum. In this case, the first term (multiplied by the weight) would be replaced by $-(w_k^{(n-1)}(s) - \alpha) \sum_a \pi_k(a|s) \log \pi_k(a|s)$, leading to a non-concave problem depending on the current sign of $-(w_k^{(n-1)}(s) - \alpha)$ at iteration $n-1$, which could also fluctuate across iterations (remember that $\alpha < 1$ and $0 < w_k^{(n-1)}(s) < 1$).

Introducing a Lagrange multiplier $\lambda_k$, we form the Lagrangian

$$
\mathcal{L}_k(s, \pi_k, \lambda_k) = \sum_a \pi_k(a|s) \left[ \log r_k^{(n-1)}(s,a) + \gamma Q^{(n-1)}(s,a) \right] - (1-\alpha) \sum_a \pi_k(a|s) \log \pi_k(a|s)
$$

$$
+ \lambda_k \left( \sum_a \pi_k(a|s) - 1 \right).
$$

Differentiating with respect to $\pi_k(a|s)$ and setting it to zero to find critical points gives

$$
\frac{\partial \mathcal{L}_k}{\partial \pi_k(a|s)} = \log r_k^{(n-1)}(s,a) + \gamma Q^{(n-1)}(s,a) - (1-\alpha)\Big( \log \pi_k(a|s) + 1 \Big) + \lambda_k = 0, \quad (38)
$$

which after rearranging becomes

$$
\log \pi_k(a|s) = \frac{\log r_k^{(n-1)}(s,a) + \gamma Q^{(n-1)}(s,a) + \lambda_k - (1-\alpha)}{1-\alpha},
$$

or equivalently,

$$
\pi_k(a|s) = \exp\left( \frac{\log r_k^{(n-1)}(s,a) + \gamma Q^{(n-1)}(s,a)}{1-\alpha} \right) \exp\left( \frac{\lambda_k - (1-\alpha)}{1-\alpha} \right).
$$

Enforcing normalization yields the closed-form softmax update

$$
\pi_k^{(n)}(a|s) = \frac{[r_k^{(n-1)}(s,a)]^{\frac{1}{1-\alpha}} e^{\frac{\gamma}{1-\alpha} Q^{(n-1)}(s,a)}}{\sum_{a'} [r_k^{(n-1)}(s,a')]^{\frac{1}{1-\alpha}} e^{\frac{\gamma}{1-\alpha} Q^{(n-1)}(s,a')}}.
$$

identical to Eq. 13.

Because $r_k^{(n-1)}(s,a) > 0$ and $Q^{(n-1)}(s,a) < \infty$, the critical point is in the simplex. Therefore $w_k^{(n)}$ is the global maximum. Further, note that the global maximum is not on the simplex boundaries. We remind the reader that the fact that $\pi^{(n)}, w^{(n)}, r^{(n)}$ are all positive for all $n$ has been shown by induction (see Theorem 2).

### A.2.2 FINDING OPTIMAL MIXTURE WEIGHTS AT ITERATION $n$

To determine the optimal weights $w_k^{(n)}(s)$, we solve the constrained maximization

$$w_k^{(n)}(s) = \arg\max_w \Big[ G\big(s;\, \pi^{(n)}, w, r^{(n-1)}\big) + \gamma \sum_{k,a} w_k(s)\, \pi_k^{(n)}(a \mid s)\, Q^{(n-1)}(s,a) \Big],$$

$$\text{subject to} \quad \sum_k w_k(s) = 1,$$

(39)

where $\pi^{(n)}$ is computed in step (2b) and $r_k^{(n-1)}(s,a) > 0$ for all $k, s, a$.

We can rewrite the objective as we saw earlier (see Eq. 26) as

$$G\big(s;\, \pi^{(n)}, w, r^{(n-1)}\big) + \gamma \sum_{k,a} w_k(s)\, \pi_k^{(n)}(a \mid s)\, Q^{(n-1)}(s,a)$$

$$= (1 - \alpha) \sum_k w_k(s) \log F_k^{(n-1)}(s) - \sum_k w_k(s) \log w_k(s) \,,$$

(40)

where

$$F_k^{(n-1)}(s) = \sum_a [r_k^{(n-1)}(s,a)]^{\frac{1}{1-\alpha}} e^{\frac{\gamma}{1-\alpha} Q^{(n-1)}(s,a)} \,.$$

(41)

We note again that the objective is strictly concave in the $w_k(s)$, and therefore once again if there is a critical point in the simplex $\sum_k w_k(s) = 1$, it must be the global maximum, as it is shown below.

To enforce the simplex constraint, we introduce a Lagrange multiplier $\lambda_w$ and form

$$\mathcal{L}\big(s, w, \lambda_w\big) = (1 - \alpha) \sum_{k=1}^K w_k(s) \log F_k^{(n-1)}(s) - \sum_{k=1}^K w_k(s) \log w_k(s) + \lambda_w \Big( \sum_{k=1}^K w_k(s) - 1 \Big).$$

(42)

Differentiating $\mathcal{L}$ with respect to $w_k(s)$ and setting the derivative to zero yields

$$-\log w_k(s) - 1 + (1 - \alpha) \log F_k^{(n-1)}(s) + \lambda_w = 0.$$

(43)

Solving for $w_k(s)$ and enforcing $\sum_k w_k(s) = 1$ gives the closed-form update

$$w_k^{(n)}(s) = \frac{\big(F_k^{(n-1)}(s)\big)^{1-\alpha}}{\sum_l \big(F_l^{(n-1)}(s)\big)^{1-\alpha}} = \frac{\Big( \sum_a [r_k^{(n-1)}(s,a)]^{\frac{1}{1-\alpha}} e^{\frac{\gamma}{1-\alpha} Q^{(n-1)}(s,a)} \Big)^{1-\alpha}}{\sum_l \Big( \sum_a [r_l^{(n-1)}(s,a)]^{\frac{1}{1-\alpha}} e^{\frac{\gamma}{1-\alpha} Q^{(n-1)}(s,a)} \Big)^{1-\alpha}} \,.$$

(44)

Because $r_k^{(n-1)}(s,a) > 0$ and $Q^{(n-1)}(s,a) < \infty$, the optimal mixture weights at iteration $n$ are in its simplex, but not on its simplex boundaries.

### A.2.3 FINDING OPTIMAL RESPONSIBILITIES AT ITERATION $n$

To solve the optimization problem

$$r^{(n)}(s,\cdot) = \arg\max_r \left( G\big(s; \pi^{(n)}, w^{(n)}, r\big) + \gamma \sum_{k,a} w_k^{(n)}(s)\, \pi_k^{(n)}(a|s) Q^{(n-1)}(s,a) \right),$$

we note that the only term involving $r$ affecting the maximization is

$$\sum_{k,a} w_k^{(n)}(s)\, \pi_k^{(n)}(a|s)\, \log r_k(s,a),$$

subject to the normalization constraint that for every state-action pair $(s,a)$ the responsibilities satisfy $\sum_{k=1} r_k(s,a) = 1$. Note that when $r_k^{(n)}(s,a) > 0$ and $\pi_k^{(n)}(a|s) > 0$ the function is strictly concave in $r$. Therefore, if there is a critical point inside the simplex $\sum_k r_k(s,a) = 1$, it must be the global maximum, as we show below.

Introducing a Lagrange multiplier $\lambda_r(a|s)$ for each state–action pair $(s, a)$, the Lagrangian can be written as

$$\mathcal{L}_r = \sum_{k=1}^{K} w_k^{(n)}(s)\, \pi_k^{(n)}(a|s) \, \log r_k(a|s) + \lambda_r(s, a) \left(1 - \sum_{k=1}^{K} r_k(s, a)\right). \tag{45}$$

Taking the derivative of the Lagrangian in equation 45 with respect to $r_k(a|s)$ yields

$$\frac{\partial \mathcal{L}_r}{\partial r_k(a|s)} = \frac{w_k^{(n)}(s)\, \pi_k^{(n)}(a|s)}{r_k(a|s)} - \lambda_r(a|s) = 0,$$

which implies

$$r_k(s, a) = \frac{w_k^{(n)}(s)\, \pi_k^{(n)}(a|s)}{\lambda_r(a|s)}.$$

Enforcing the normalization constraint

$$\sum_{k=1}^{K} r_k(a|s) = 1 \quad \implies \quad \lambda_r(a|s) = \sum_{k=1}^{K} w_k^{(n)}(s)\, \pi_k^{(n)}(a|s) = \pi_m^{(n)}(a|s),$$

we obtain the updated responsibilities

$$r_k^{(n)}(a|s) = \frac{w_k^{(n)}(s)\, \pi_k^{(n)}(a|s)}{\pi_m^{(n)}(a|s)} \ .$$

These probabilities over $k$ are the global maximum, and they do not lie on the simplex boundaries because $r_k^{(n)} > 0$ and $\pi_k^{(n)}(a|s) > 0$ (see Theorem 2).

### A.3 Unsupervised Policy Gradient Reparameterization for Continuous Control

In this section, we present the proof of Theorem 3, in which the performance measure is defined as the value of the initial state of the episode, $J(\theta) = V(s_0)$. However, the derivation generalizes straightforwardly to the case where the initial state is drawn from a distribution $\rho(s_0)$, in which case the objective becomes $J(\theta) = \mathbb{E}_{s_0 \sim \rho}[V(s_0)]$, and the gradient is taken with respect to this expectation. Our derivation is a simple generalization of the policy gradient theorem of Sutton & Barto (1998) by considering a mixture policy and incorporating, as intrinsic reward, a linear combination of the entropy of the mixture and the entropies of its component policies.

**Theorem 3.** *Let $\pi_{m,\theta}(a|s)$ be a mixture policy over continuous actions, and define the state-value function as in Eq. 3. Then the gradient of the performance objective $J(\theta) = V(s_0)$ admits the form*

$$\nabla_\theta J(\theta) = \int_{s,a} d_{\pi_{m,\theta}}(s)\pi_{m,\theta}(a|s)\Big(\nabla_\theta \log \pi_{m,\theta}(a|s)Q(s,a) + \nabla_\theta R(s; \pi_\theta, w_\theta)\Big) \, da \, ds.$$

*where* $\nabla_\theta R(s; \pi_\theta, w_\theta) = \nabla_\theta \mathcal{H}\big(\pi_{m,\theta}(\cdot|s)\big) - \alpha \nabla_\theta \sum_{k=1}^{K} w_{k,\theta}(s)\, \mathcal{H}(\pi_{k,\theta}(\cdot|s)), \quad \pi_\theta = \{\pi_{1,\theta}, ..., \pi_{K,\theta}\}$ *and* $w_\theta = \{w_{1,\theta}, ..., w_{K,\theta}\}$

*Proof.* The gradient of the state-value function can be written in terms of the action-value function as

$\nabla_\theta V(s)$

$= \nabla_\theta R(s; \pi_\theta, w_\theta) + \nabla_\theta \gamma \int_a \pi_{m,\theta}(a|s) Q(s,a) \, da$

$= \nabla_\theta R(s; \pi_\theta, w_\theta) + \gamma \int_a \nabla_\theta \pi_{m,\theta}(a|s) Q(s,a) \, da + \gamma \int_a \pi_{m,\theta}(a|s) \nabla_\theta Q(s,a) \, da$

$= \nabla_\theta R(s; \pi_\theta, w_\theta) + \gamma \int_a \nabla_\theta \pi_{m,\theta}(a|s) Q(s,a) \, da + \gamma \int_a \pi_{m,\theta}(a|s) \nabla_\theta \int_{s'} p(s'|s,a) V(s') \, ds \, da$

$= \nabla_\theta R(s; \pi_\theta, w_\theta) + \gamma \int_a \nabla_\theta \pi_{m,\theta}(a|s) Q(s,a) \, da + \gamma \int_{a,s'} \pi_{m,\theta}(a|s) p(s'|s,a) \nabla_\theta V(s') \, ds \, da$

$= \nabla_\theta R(s; \pi_\theta, w_\theta) + \gamma \int_a \nabla_\theta \pi_{m,\theta}(a|s) Q(s,a) \, da +$

$+ \gamma \int_{a,s'} \pi_{m,\theta}(a|s) p(s'|s,a)$

$\left( \nabla_\theta R(s'; \pi_\theta, w_\theta) + \int_{a'} \nabla_\theta \pi_{m,\theta}(a'|s') Q(s',a') \, da' + \gamma \int_{a',s''} \pi_{m,\theta}(a'|s') p(s''|s',a') \nabla_\theta V(s'') \, ds' \, da' \right)$

$= \int_x \sum_t^\infty Pr(s \to x, t, \pi_{m,\theta}) \left( \nabla_\theta R(s; \pi_\theta, w_\theta) + \int_a \nabla_\theta \pi_{m,\theta}(a|x) Q(x,a) \right)$

$\hfill (46)$

where we define $Pr(s \to x, t, \pi_\theta)$ similar to Sutton & Barto (1998) as the discounted probability of transitioning from state s to state x in t steps under the mixture policy $\pi_{m,\theta}$. Then we can find that

$\nabla_\theta J(\theta) = \nabla_\theta V(s_0)$

$= \int_s \sum_t^\infty Pr(s_0 \to s, t, \pi_{m,\theta}) \left( \nabla_\theta R(s; \pi_\theta, w_\theta) + \int_a \nabla_\theta \pi_{m,\theta}(a|s) Q(s,a) \right) \quad (47)$

$= \int_{s,a} d_{\pi_{m,\theta}}(s) \pi_{m,\theta}(a|s) \left( \nabla_\theta \log \pi_{m,\theta}(a|s) Q(s,a) + \nabla_\theta R(s; \pi_\theta, w_\theta) \right)$

$\hfill \square$

In this work, we reparametrize our component policies as $\pi_{k,\theta}(a|s) = p(\varepsilon)$ where $a = f_\theta(\varepsilon; s, k)$ and $p(\cdot)$ is the noise distribution. Here we prove theorem 4 (see Sec. 3) in which we provide an unbiased reparametrization of our component policies.

**Theorem 4.** *Let $\hat{f}_\theta = f_\theta(\varepsilon; s, k)$ then the gradients of objective function under our reparametrization become*

$\nabla_\theta J(\pi_{m,\theta})$

$= \mathbb{E}_{\substack{s \sim d_{\pi_{m,\theta}} \\ k \sim Cat(w_{k,\theta}(s)) \\ \varepsilon \sim p}} \left[ \nabla_\theta \log w_{k,\theta}(s) \left( Q_{\pi_{m,\theta}}(s, \hat{f}_\theta) - \log \pi_{m,\theta}(\hat{f}_\theta|s) + \alpha \log \pi_{k,\theta}(\hat{f}_\theta|s) \right) \right.$

$\hfill (48)$

$\left. + \nabla_\theta \hat{f}_\theta \, \nabla_a \left( Q_{\pi_{m,\theta}}(s, a) - \log \pi_{m,\theta}(a|s) + \alpha \log \pi_{k,\theta}(a|s) \right) \Big|_{a = \hat{f}_\theta} \right],$

*where $d_{\pi_{m,\theta}}(s)$ is the discounted occupancy measure under $\pi_{m,\theta}$.*

*Proof.* Our proof depends heavily on a similar proof provided by He et al. (2025), where they discuss the reparametrization trick for entropy regularized policy gradient theorem for mixture policies. The only difference between our case and their case is the weighted sum of component entropy and the absence of the extrensic reward from the environment. We start with the extended policy gradient Theorem 3 which states that

$\nabla_\theta J(\theta) = \int_{s,a} d_{\pi_{m,\theta}}(s) \pi_{m,\theta}(a|s) \left( \nabla_\theta \log \pi_{m,\theta}(a|s) Q(s,a) + \nabla_\theta R(s; \pi_\theta, w_\theta) \right) \, da \, ds. \quad (49)$

We can break this last equation into 3 parts that we can handle separately. The first part is for the state-action value function which recovers the vanilla policy gradient theorem (Sutton & Barto, 1998). He et al. (2025) found that

$$
\int_{s,a} d_{\pi_{m,\theta}}(s)\pi_{m,\theta}(a|s)\nabla_\theta \log \pi_{m,\theta}(a|s)Q_{\pi_{m,\theta}}(s,a)\, da\, ds
$$

$$
= \mathbb{E}_{\substack{s\sim d_{\pi_{m,\theta}}\\ k\sim\mathrm{Cat}(w_{k,\theta}(s))\\ \epsilon\sim p(\epsilon)}}\Big[Q_{\pi_{m,\theta}}\big(s,\hat{f}_\theta\big)\,\nabla_\theta \log w_{k,\theta}(s) + \nabla_\theta \hat{f}_\theta\,\nabla_a Q_{\pi_{m,\theta}}(s,a)\big|_{a=\hat{f}_\theta}\Big]. \tag{50}
$$

The second part handles the entropy of the mixture policy and similarly He et al. (2025) found that:

$$
\int_{s,a} d_{\pi_{m,\theta}}(s)\,\pi_{m,\theta}(a\mid s)\,\nabla_\theta \mathcal{H}\big(\pi_{m,\theta}(\cdot\mid s)\big)\, da\, ds \tag{51}
$$

$$
= -\mathbb{E}_{\substack{s\sim d_{\pi_{m,\theta}}\\ k\sim\mathrm{Cat}(w_{k,\theta}(s))\\ \epsilon\sim p(\epsilon)}}\Big[\log \pi_{m,\theta}\big(\hat{f}_\theta\mid s\big)\,\nabla_\theta \log w_{k,\theta}(s) + \nabla_\theta \hat{f}_\theta\,\nabla_a \log \pi_{m,\theta}(a\mid s)\big|_{a=\hat{f}_\theta}\Big]. \tag{52}
$$

The third term is

$$
\int_{s,a} d_{\pi_{m,\theta}}(s)\pi_\theta(a|s)\nabla_\theta\left(\sum_k w_{k,\theta}(s)\mathcal{H}(\pi_{k,\theta}(\cdot|s))\right)da\, ds\,. \tag{53}
$$

Note first that we can rewrite the entropy of component k with reparametrization as:

$$
\mathcal{H}(\pi_{k,\theta}(\cdot\mid s)) = -\mathbb{E}_{\epsilon\sim p(\epsilon)}\Big[\log \pi_{k,\theta}\big(\hat{f}_\theta\mid s\big)\Big]. \tag{54}
$$

Then the gradient

$$
\sum_k w_{k,\theta}(s)\nabla_\theta\mathcal{H}(\pi_{k,\theta}) = -\sum_k w_{k,\theta}(s)\nabla_\theta\mathbb{E}_\epsilon\big[\log \pi_{k,\theta}(\hat{f}_\theta\mid s)\big]
$$

$$
= -\mathbb{E}_{\substack{k\sim\mathrm{Cat}(w_{k,\theta}(s))\\ \epsilon\sim p(\epsilon)}}\Big[\nabla_\theta \hat{f}_\theta\,\nabla_a \log \pi_{k,\theta}(a\mid s)\big|_{a=\hat{f}_\theta}\Big]. \tag{55}
$$

In addition we have

$$
\sum_k \nabla_\theta w_{k,\theta}(s)\,\mathcal{H}(\pi_k(\cdot|s)) = -\sum_k w_{k,\theta}(s)\nabla_\theta \log w_{k,\theta}(s)\mathbb{E}_\epsilon\big[\log \pi_{k,\theta}(\hat{f}_\theta\mid s)\big]
$$

$$
= -\mathbb{E}_{\substack{k\sim\mathrm{Cat}(w_{k,\theta}(s))\\ \epsilon\sim p(\epsilon)}}\big[\nabla_\theta \log w_{k,\theta}(s)\log \pi_k(\hat{f}_\theta\mid s)\big] \tag{56}
$$

$$
\int_{s,a} d_{\pi_{m,\theta}}(s)\pi_{m,\theta}(a|s)\nabla_\theta\left(\sum_k w_{k,\theta}(s)\mathcal{H}(\pi_{k,\theta_\theta}(\cdot|s))\right)da\, ds\,.
$$

$$
= -\mathbb{E}_{\substack{s\sim d_{\pi_{m,\theta}}\\ kk\sim\mathrm{Cat}(w_{k,\theta}(s))\\ \epsilon\sim p(\epsilon)}}\Big[\nabla_\theta \log w_{k,\theta}(s)\,\log \pi_k\big(\hat{f}_\theta\mid s\big) + \nabla_\theta \hat{f}_\theta\,\nabla_a \log \pi_k(a\mid s)\big|_{a=\hat{f}_\theta}\Big].
$$

Combining these 3 parts with multiplying the last one with $-\alpha$ completes our proof.

$\square$

## A.4    Mode-Based vs. Stochastic Component Execution

The behavioral tokens in our method are the component policies $\pi_k(\cdot\mid s)_{k=1}^K$ themselves; the discrete high-level controller selects a token index $k\in 1,\ldots,K$ at each timestep. To execute this selection as a continuous control signal, there are two natural grounding strategies: either sample an action $a_t\sim\pi_k(\cdot\mid s_t)$ from the selected component, or execute the deterministic mode $a_t=\mu_k(s_t)$ of that component. In all experiments reported in the main text we adopt the mode-based approach for three practical reasons. First, using modes yields a one-to-one mapping $(s,k)\mapsto a$ and thereby

induces a standard discrete-action MDP, which allows us to directly employ off-the-shelf discrete RL algorithms (DQN, discrete PPO) and enables fair comparison with random quantization and unsupervised skill discovery methods which are also applied deterministically. Second, using modes removes the additional stochasticity that would arise from sampling $\pi_k(\cdot \mid s)$, reducing variance in value estimation and isolating the effect of the learned token set rather than conflating it with per-step sampling noise. Third, our maxi-mix-mini-com objective explicitly drives each component $\pi_k(\cdot \mid s)$ toward low entropy and strong specialization via the $-\alpha \sum_k w_k(s_t)\mathcal{H}(\pi_k(\cdot|s_t))$ penalty (Eq. 2); in this low-entropy regime the mode $\mu_k(s)$ is a faithful summary of the component's behavior.

Importantly, nothing in our framework prevents us from using the stochastic variant—i.e., having the discrete policy select $k$ and then sampling $a_t \sim \pi_k(\cdot \mid s_t)$ for execution. This simply corresponds to a hierarchical policy over one-step stochastic skills rather than over deterministic quantized actions, and may be preferable in domains where preserving exploration noise at the low level is beneficial. We leave systematic comparison of these two grounding strategies to future work.

### A.5 Implementation Details of the Policy Iteration Algorithm

To evaluate our policy iteration algorithm, we employed a simple grid-world environment comprising three cell types: walls (impassable), terminal states, and feasible (navigable) states. The agent's objective is to maximize the entropy of its state–action distribution; consequently, it must avoid regions with constrained movement. By definition, actions that would lead into wall cells are excluded from the agent's action set. Likewise, states adjacent to terminal cells are implicitly disfavored, since transitions into a terminal state immediately terminate an episode and prevent further entropy accumulation. As shown in Fig. 4, the highest-probability actions drive the agent toward the center of the upper-right room, where it remains maximally distant from both walls and terminal states, thus preserving the greatest potential for entropy maximization.

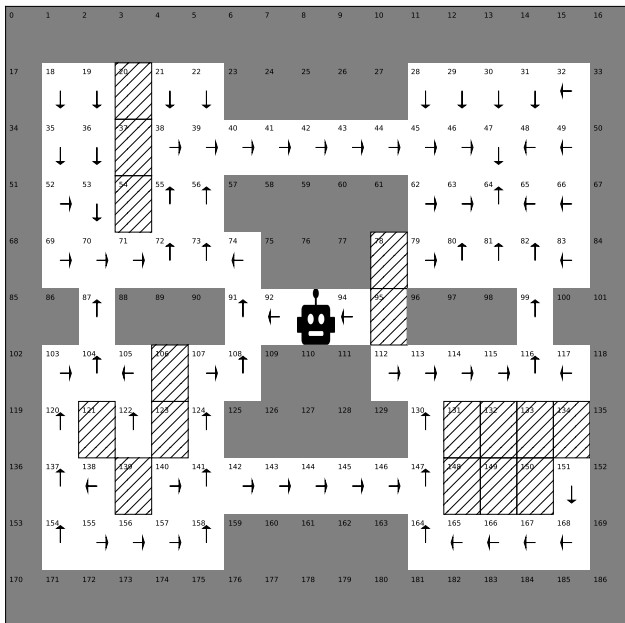

Figure 4: Grid-world environment. Grey cells denote impassable walls, and hatched cells indicate terminal states. Each cell is labeled by its state index. The robot icon marks the agent's initial state. Although the optimal mixture policy is stochastic, arrows show the action with the highest selection probability at each state.

An immediate concern is whether the same behavior would collapse to a uniform distribution if every state, including those adjacent to walls or terminal cells, were forced to retain the full action set. Figures 5.a and 5.b reproduce the analysis of state 141 from Fig. 4 with four- and three-component mixtures, respectively, while now enforcing all actions even near walls; despite this imposed symmetry, both mixtures still shift probability mass away from moves that approach terminal states. Figure 5.c

shows the solution for the grid world in Fig. 4 under the same constraint. The resulting mixture remains structured: actions that lead back toward terminal regions are systematically downweighted rather than averaged out. Consequently, even when admissible action sets are equalized across states, the entropy-seeking policy does not collapse to a uniform distribution, but instead continues to favor trajectories that prolong future entropy accumulation.

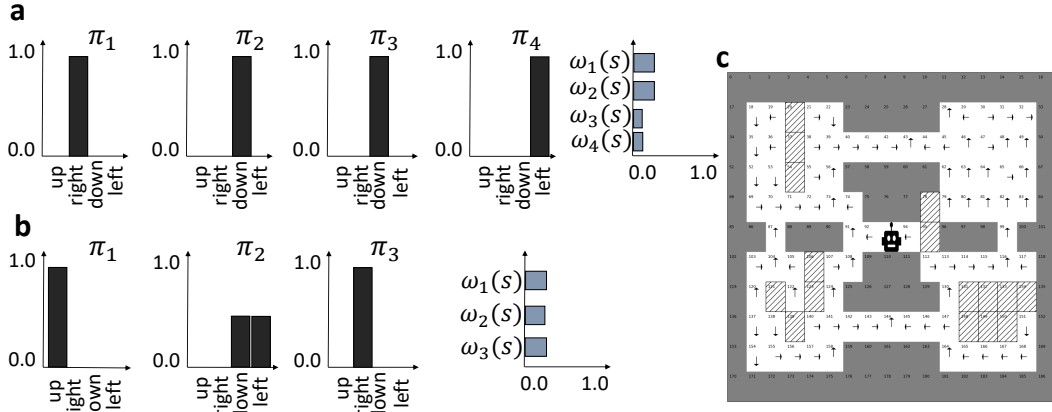

Figure 5: Effect of enforcing identical action sets for wall-adjacent states. (a,b) repeats the analysis of state 141 with four- and three-component mixtures, respectively, while retaining all actions even near walls; the resulting distributions remain non-uniform and favor moves away from terminal cells. (c) presents the mixture policy—constructed as in the example of Fig. 4 but with all actions retained even near walls—and shows that the resulting distribution maintains a structured preference that steers the agent away from terminal regions.

## A.6 IMPLEMENTATION DETAILS

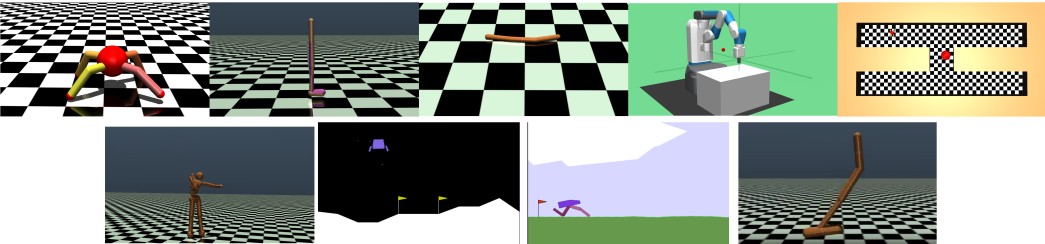

Figure 6: Benchmark environments used to evaluate our framework. From left to right: Ant-v5, Walker2D-v5, Swimmer-v5, FetchReach-v2, AntMaze, Humanoid-v5, BipedalWalker-v3, LunarLanderContinuous-v3 and Hopper-v5. These environments span diverse locomotion and manipulation tasks with varying dynamics and control complexity.

**Unsupervised Learning**: In this section, we detail the implementation of our algorithm for learning mixture components in the continuous action domain. The mixture policy is parameterized by a single neural network with $2K + 1$ output heads operating on an $M$-dimensional action space. Specifically, $K$ of these heads produce the component means $\mu_i \in \mathbb{R}^M$, another $K$ produce the log–standard deviations $\log \sigma_i \in \mathbb{R}^M$, and the final head produces unnormalized mixing logits $\hat{w} \in \mathbb{R}^K$, from which mixing weights $w \in [0,1]^K$ are obtained using the Gumbel–Softmax trick, ensuring $\sum_{i=1}^{K} w_i = 1$. The log–standard deviations are exponentiated to obtain positive standard deviations for the Gaussian components, which are then used to form $\mathrm{Normal}(\mu_i, \sigma_i^2)$ distributions.

All component parameters are computed concurrently through a shared feature extractor, with each head responsible for its respective output. Training was conducted to solve the problem defined in Eq. 2. We ran experiments for Swimmer-v5 and FetchReach-v2 for 100,000 environment steps, for Ant-v5 for 200,000 environment steps, and for Humanoid-v5 and Walker2D-v5 for 2,000,000 environment steps, using a discount factor $\gamma = 0.9$ to bias learning toward short-term diversity rather than long-horizon planning. Empirical evaluation demonstrates that this discount factor yields faster convergence to diverse behaviors within the intended temporal scale. To prevent degenerate policies in Ant-v5—where the agent could fall, flip onto its back, and generate high-entropy leg movements in mid-air—we terminate each episode whenever the torso contacts the ground. This termination rule was applied in both the unsupervised variant and the reward-accumulating maze variant, as well as during the four-directional evaluation for our agent and all baselines. We apply the episode termination conditions in the Humanoid-v5 and Walker2D-v5 environments exactly as specified in their respective definitions. In contrast, the Swimmer and FetchReach (robot) environments were evaluated without any episode termination conditions. See our hyperparameters in Table 7. For further details, see Towers et al. (2024).

**Downstream Tasks**: For the results in Table 1, where we test the reusability of our behavioral tokens, all methods are trained for $1 \times 10^7$ environment steps and evaluated over 10 random seeds. To ensure a fair comparison, health bonuses and control costs are omitted from the reward function. We assess performance on five locomotion tasks: maximizing torso velocity along the $+x$, $-x$, $+y$, and $-y$ axes (directional locomotion), and reaching a central goal in a maze from a random perimeter start (maze navigation; see Fig. 6). Continuous baselines are trained only on the $+x$ task (using symmetry for other directions), while our method is trained separately on each direction and averaged. For every Ant experiment (Table 1), we pretrain the bank of behavioral tokens once in the standard, open Ant arena (without maze walls) using only the agent's proprioceptive observation vector $s_t$ comprising joint angles, joint velocities, and body-centric features. Downstream directional controllers therefore act in the same open layout, whereas the maze-navigation controller encounters a new layout with walls; in both cases the pretrained components remain frozen. The discrete controller always receives the proprioceptive state $s_t$, and only the maze task augments that input with achieved and desired goals $(s_t, g_t^{\mathrm{ach}}, g^{\mathrm{des}})$ following the standard goal-conditioned RL formulation. In contrast, the low-level components never observe goal variables or layout-specific signals and continue to act solely on $s_t$. For the results in 2, we use the official implementation of Park et al. (2024) to train the unsupervised skill discovery (USD) method after omitting the $x$ and $y$ coordinates from the state space. All USD methods were trained for $3 \times 10^6$ steps. We then trained discrete PPO

(Schulman et al., 2017) on top of our components and on the USD methods for $1 \times 10^7$ steps. The same configuration is used for the results in 6, except that for FetchReach-v2 we use DQN (Mnih et al., 2013) due to its empirically superior performance on that environment. Thus, Ant, Humanoid, Swimmer, Walker2D, Hopper, Bipedal, LunarLander and AntMaze all employ discrete PPO as the high-level controller, whereas FetchReach-v2 uses DQN.

**Compute Resources and Experimental Setup:** We document the computational resources and environment used for all experiments, including both unsupervised pretraining and downstream evaluation. Each batch of five agents was trained on a cluster node featuring an NVIDIA T4 GPU, 16 CPU cores, and 4 GB of RAM per core. The CPUs included models such as Intel Xeon and Broadwell. Training time per batch ranged from 10 to 50 hours, while baseline models required less time. All experiments were executed within a Docker container based on the `nvidia/cuda:12.0.0-base-ubuntu20.04` image. Simulations were run using MuJoCo v3.1.6, with full configuration specified in the provided Dockerfile. GPU memory usage during training typically ranged from 8 GB to 10 GB. This setup ensures consistent dependency management and supports full reproducibility of the reported results.

**Hyperparameter Selection Protocol** All training runs share the same optimizer, learning rate, batch construction, and network architecture listed in Table 7; these values are fixed across every environment. The only method-specific degrees of freedom for our mixture policy are the specialization weight $\alpha$ and the number of components $K$. Rather than tuning $(\alpha, K)$ on a per-environment basis, we evaluate a small grid of combinations and report the associated performance and diversity statistics (e.g., Tables 1 and 6).

For both tiers of downstream RL algorithms, we use the Stable-Baselines3 implementations (Raffin et al., 2021): SAC and continuous PPO serve as continuous-action baselines, while discrete PPO and DQN act as the high-level controllers over our discrete behavioral token banks. Their default hyperparameters—network width/depth, training budget, discount, and optimizer schedule—are applied uniformly to every environment and are never retuned when we swap in our components versus random action alphabets or unsupervised skill baselines. Likewise, the unsupervised skill-discovery baselines rely on the official METRA release of Park et al. (2024), including their recommended hyperparameters, without additional sweeps. Consequently, no method—ours or any baseline—receives environment-specific hyperparameter advantages; differences arise solely from the learned tokens sets themselves.

## A.7 ADDITIONAL EXPERIMENTAL RESULTS

**Learning dynamics of discrete vs. continuous controllers.** Figure 7 plots episodic return as a function of environment interactions for our method with $K = 16$ and $K = 64$ components alongside SAC and PPO on the same continuous-control benchmarks used in Table 3. For each method, we plot the mean over 10 random seeds and show 95% confidence intervals, where individual seed curves (except for FetchReach-v2) are first exponentially smoothed with coefficient $\alpha = 0.9$. Across all environments, SAC attains the highest asymptotic returns, with PPO typically slightly below. Our discrete controller, which operates on a fixed bank of unsupervised components, exhibits stable learning behavior and converges to a significant fraction of the SAC/PPO performance on all tasks, with larger $K$ generally improving final returns. These results support the claim that a modest number of learned behavioral components is sufficient to support strong downstream control in a variety of continuous-control benchmarks, even though we do not aim to surpass SAC or PPO in single-task sample efficiency.

**Sample efficiency in a stabilization-constrained Ant variant.** To further assess the practical benefits of our approach in settings where stabilization is genuinely challenging, we design a modified Ant-v5 environment that mirrors the "do not fall" constraint in a controlled way. Concretely, we constrain the torso height to a narrow healthy band $z \in [0.6, 0.7]$, compared to the standard MuJoCo band $[0.3, 1.0]$; episodes terminate immediately upon leaving this band, which substantially tightens the stabilization requirement (all episodes start at $z = 0.65$). The downstream reward is defined purely as forward velocity, with no survival bonus, to avoid giving our pretrained tokens an automatic advantage. We first pretrain a mixture policy using only the maxi–mix–mini–com objective for 500,000 interaction steps under this termination rule, then freeze the components and train a high-level controller for 200,000 steps on the forward-only task; SAC and PPO are trained from scratch

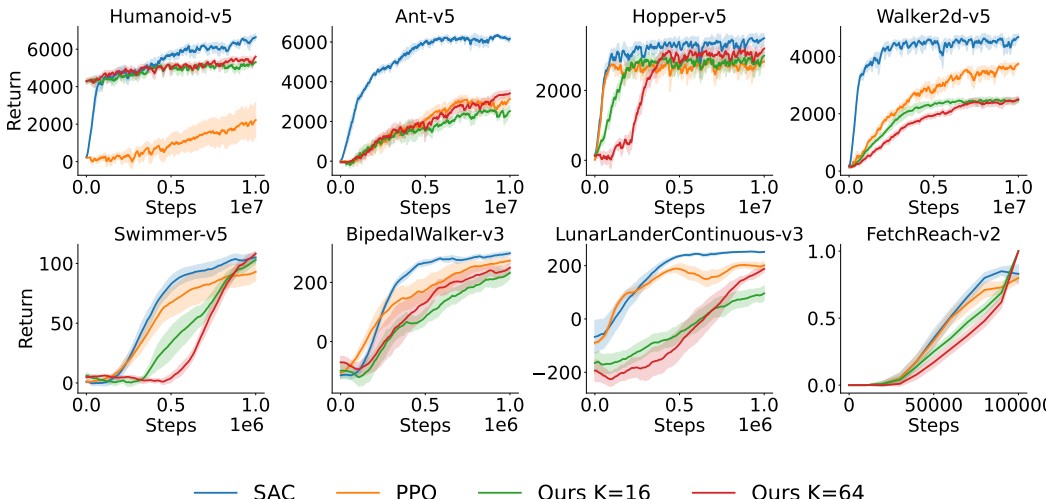

Figure 7: Learning curves for our method with $K \in \{16, 64\}$ behavioral components compared to continuous-action baselines SAC and PPO on the continuous-control benchmarks reported in Table 3. For each method, curves show the mean episodic return over 10 random seeds, and shaded regions indicate 95% confidence interval. Individual seed curves (except for FetchReach-v2) are exponentially smoothed with coefficient $\alpha = 0.9$ for visual clarity. Despite acting through a fixed, low-cardinality discrete action set, our method converges to a substantial fraction of the final returns obtained by the continuous-action baselines on all tasks.

for 200,000 steps on the same task. Our method achieves a return of $55.81 \pm 6.62$ with mean episode length $498.8 \pm 102.67$ steps, whereas SAC attains $9.41 \pm 3.30$ with length $8.2 \pm 1.92$, and PPO $5.32 \pm 5.21$ with length $7.4 \pm 2.88$. Thus, for the same downstream training budget, our pretrained tokens yield more than a fivefold improvement in average return over SAC and PPO and produce policies that survive for hundreds of steps. This behavior arises because maximizing entropy under the narrow healthy band forces the mixture components to maintain upright, stable dynamics over long horizons while spanning diverse actions within the safe region, whereas SAC/PPO trained from scratch remain in a near-immediate-fall regime with very short episodes and limited usable learning signal. Taken together with the Humanoid results, these findings indicate that, in environments where failure terminates behavior and stabilization is difficult, behavioral-token pretraining confers a tangible sample-efficiency and robustness advantage on top of the compact, transferable discrete action interface established in our other experiments.

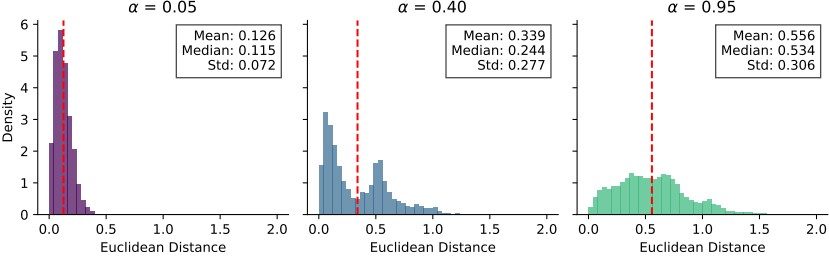

Figure 8: Pairwise Euclidean distance densities between component action means for $\alpha = 0.05, 0.4$, and $0.95$.

**Effect of entropy regularization on component spacing** Subsequently, we tested the maxi-mix-mini-com entropy objective on the MuJoCo Ant benchmark (Todorov et al., 2012). For three values of the regularization coefficient ($\alpha = 0.05$, $0.4$, $0.95$), we trained a 16-component Gaussian mixture policy and measured all pairwise Euclidean distances between the component mean action vectors under deterministic execution. Figure 8 shows the resulting pairwise distance densities. As $\alpha$ increases, the

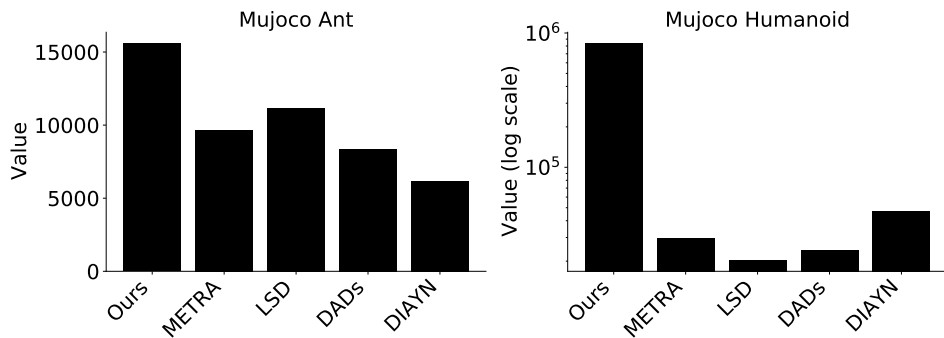

Figure 9: Within-mixture component divergence (maxi-mix-mini-com) vs. skill divergence (skill discovery methods), measured as the average sum of pairwise Jeffreys divergences per state over 20k evaluations (10k entropy-driven (Ramírez-Ruiz et al., 2024) + 10k uniform) on Ant-v5 and Humanoid-v5; ours shown with $\alpha = 0.8$ (Ant) and $\alpha = 1.0$ (Humanoid).

average distance between components grows, the variation in those distances becomes larger, and the median distance shifts upward, demonstrating that components move farther apart on average and with greater spread. At the same time, each density curve retains density near zero, indicating that many component pairs remain closely clustered even at higher $\alpha$. Together, these trends show that raising $\alpha$ both expands the policy's overall span of the action space and preserves small groups of highly similar, specialized components.

**Measuring within-mixture diversity.** We seek to quantify how diverse our behavioral tokens are compared to the diversity between skills learned by skill discovery methods, since greater separation yields a richer bank of behavioral tokens (quantized actions) for composition. Concretely, for our approach we compute, at each state, the sum of pairwise Jeffreys (symmetrized KL) divergences between the components $\{\pi_k(\cdot \mid s)\}$ of the same mixture policy $\pi_m(a \mid s)$; for each skill discovery method, we analogously compute divergences between its skills $\pi(\cdot \mid s, z)$ across latent codes $z$. We average this per-state quantity over 20,000 evaluation states assembled from two exploration regimes: (i) an entropy-driven regime using MOP (Ramírez-Ruiz et al., 2024), where MOP is trained to maximize the agent's policy entropy for $1 \times 10^6$ environment steps and then executed to collect trajectories until $10k$ states are obtained; and (ii) a uniform-action regime in which actions are sampled uniformly at random from the environment action space and executed to collect $10k$ states (no training). We report the resulting means on MuJoCo Ant-v5 and Humanoid-v5 (Fig. 9). In the configurations shown ($\alpha=0.8$ for Ant; $\alpha=1.0$ for Humanoid), our mixture attains the largest average pairwise Jeffreys divergence (Fig. 9), indicating more distinct behavioral tokens within the mixture. Supplementary videos (see Videos 5) illustrate that the mixture components exhibit greater behavioral diversity and yield a broader behavioral repertoire with more effective actions.

In addition, we report the mean pairwise Euclidean distance (MPED) for various combinations of $\alpha$ and $K$ in Ant-v5, averaged over 5 random seeds. The results indicate that both increasing $\alpha$ and $K$ increasing lead to higher MPED values, reflecting greater diversity among components. Notably, the effect of $\alpha$ is more pronounced when $K$ is small, as components must spread more aggressively to achieve higher overall entropy. In contrast, for larger $K$, diversity is already induced by the increased number of components, each of which must occupy a distinct region of the action space to maintain low entropy while contributing to the overall mixture entropy.

Table 4: Mean Pairwise Euclidean Distance for Various $\alpha$ and $K$ Values for Ant-v5.

| $K$ | $\alpha = 0.10$ | $\alpha = 0.20$ | $\alpha = 0.30$ |
|---|---|---|---|
| 16 | 0.58 | 0.96 | 1.32 |
| 64 | 1.24 | 1.30 | 1.33 |
| 128 | 1.33 | 1.34 | 1.35 |

Table 5: Performance (mean $\pm$ SE) for Humanoid, Walker2D, FetchReach and Swimmer under different action-set configurations with $K = 4$, 16, and 64 components. "m" uses the discrete alphabet $\{-m, 0, +m\}$ to sample each action dimension independently.

| Env. | Method | $K = 4$ | $K = 16$ | $K = 64$ |
|------|--------|---------|----------|----------|
| FetchReach-v2 | SAC | | $0.84 \pm 0.07$ | |
| | Ours | $0.460 \pm 0.04$ | $1.000 \pm 0.000$ | $1.000 \pm 0.000$ |
| | $\mathcal{U}(-1.0, 1.0)$ | $0.16 \pm 0.09$ | $0.96 \pm 0.02$ | $1.000 \pm 0.000$ |
| | $\mathcal{N}(\mu = 0, \sigma = 0.2)$ | $0.14 \pm 0.09$ | $0.76 \pm 0.06$ | $0.88 \pm 0.04$ |
| | $m = 0.1$ | $0.180 \pm 0.050$ | $0.360 \pm 0.051$ | $0.140 \pm 0.060$ |
| | $m = 0.2$ | $0.080 \pm 0.006$ | $0.540 \pm 0.125$ | $0.620 \pm 0.139$ |
| | $m = 0.5$ | $0.320 \pm 0.013$ | $0.940 \pm 0.060$ | $1.000 \pm 0.000$ |
| Swimmer | SAC | | $110.54 \pm 1.52$ | |
| | Ours | $89.41 \pm 2.11$ | $113.77 \pm 3.32$ | $117.57 \pm 1.17$ |
| | $\mathcal{U}(-1.0, 1.0)$ | $77.50 \pm 1.31$ | $103.87 \pm 10.78$ | $111.33 \pm 7.54$ |
| | $\mathcal{N}(\mu = 0, \sigma = 0.2)$ | $44.36 \pm 2.28$ | $86.73 \pm 3.79$ | $42.60 \pm 3.35$ |
| | $m = 0.1$ | $34.99 \pm 4.22$ | $46.50 \pm 2.12$ | $48.52 \pm 2.10$ |
| | $m = 0.2$ | $47.25 \pm 2.84$ | $45.23 \pm 6.64$ | $54.52 \pm 2.60$ |
| | $m = 0.5$ | $30.56 \pm 4.72$ | $83.59 \pm 5.05$ | $76.08 \pm 3.86$ |
| Walker2D | SAC | | $4844.77 \pm 132.30$ | |
| | Ours | $1479.30 \pm 44.76$ | $2449.72 \pm 34.76$ | $2407.30 \pm 112.09$ |
| | $\mathcal{U}(-1.0, 1.0)$ | $11.52 \pm 0.87$ | $2031.73 \pm 367.35$ | $1434.98 \pm 296.94$ |
| | $\mathcal{N}(\mu = 0, \sigma = 0.2)$ | $351.02 \pm 42.17$ | $369.71 \pm 048.80$ | $486.66 \pm 39.71$ |
| | $m = 0.1$ | $289.02 \pm 53.23$ | $357.84 \pm 30.44$ | $466.15 \pm 40.78$ |
| | $m = 0.2$ | $296.34 \pm 69.01$ | $748.64 \pm 120.44$ | $485.64 \pm 40.44$ |
| | $m = 0.5$ | $1016.90 \pm 250.58$ | $447.81 \pm 78.24$ | $1995.08 \pm 57.28$ |
| | $m = 1.0$ | $438.46 \pm 28.89$ | $1676.77 \pm 35.62$ | $749.45 \pm 55.59$ |
| Humanoid | SAC | | $6846.10 \pm 42.00$ | |
| | Ours | $5092.87 \pm 42.45$ | $5407.02 \pm 120.00$ | $5498.92 \pm 80.03$ |
| | $\mathcal{U}(-0.4, 0.4)$ | $525.78 \pm 62.45$ | $512.46 \pm 41.04$ | $610.24 \pm 51.57$ |
| | $\mathcal{N}(\mu = 0, \sigma = 0.2)$ | $418.04 \pm 11.74$ | $522.18 \pm 07.14$ | $508.72 \pm 41.10$ |
| | $m = 0.1$ | $487.29 \pm 29.88$ | $467.53 \pm 28.71$ | $544.66 \pm 30.21$ |
| | $m = 0.2$ | $561.22 \pm 33.12$ | $608.85 \pm 10.29$ | $583.90 \pm 20.21$ |
| | $m = 0.4$ | $415.37 \pm 18.92$ | $489.64 \pm 22.15$ | $503.28 \pm 15.47$ |

Table 6: Ant-v5: returns after 10M steps. "Uniform" draws each action dimension from $[-1, 1]$. "m" uses the discrete alphabet $\{-m, 0, +m\}$ to sample each action dimension independently.

| Method | Parameter | 16 | 64 | 128 |
|--------|-----------|-----|-----|-----|
| SAC | – | | $6115.90 \pm 40.61$ | |
| Ours | $\alpha = 0.1$ | $1669.40 \pm 22.22$ | $2399.07 \pm 18.95$ | $3726.89 \pm 32.66$ |
| | $\alpha = 0.2$ | $2936.96 \pm 23.18$ | $3580.79 \pm 31.87$ | $3399.25 \pm 20.05$ |
| | $\alpha = 0.3$ | $1864.21 \pm 17.72$ | $3280.89 \pm 23.05$ | $3538.76 \pm 31.33$ |
| Random Set of Actions | $m = 1.0$ | $2677.58 \pm 281.31$ | $1287.32 \pm 435.12$ | $1516.34 \pm 125.42$ |
| | $m = 0.5$ | $2339.68 \pm 202.66$ | $3198.97 \pm 397.03$ | $1783.20 \pm 096.99$ |
| | $m = 0.2$ | $1883.51 \pm 264.06$ | $1751.37 \pm 255.22$ | $1769.44 \pm 292.19$ |
| | $m = 0.1$ | $1044.44 \pm 081.46$ | $1059.06 \pm 156.44$ | $0955.92 \pm 174.86$ |
| | $\mathcal{U}(-1.0, 1.0)$ | $2698.80 \pm 372.46$ | $1541.63 \pm 196.43$ | $1101.21 \pm 206.97$ |
| | $\mathcal{N}(\mu = 0, \sigma = 0.2)$ | $1841.58 \pm 060.78$ | $2470.82 \pm 120.45$ | $2335.90 \pm 180.36$ |

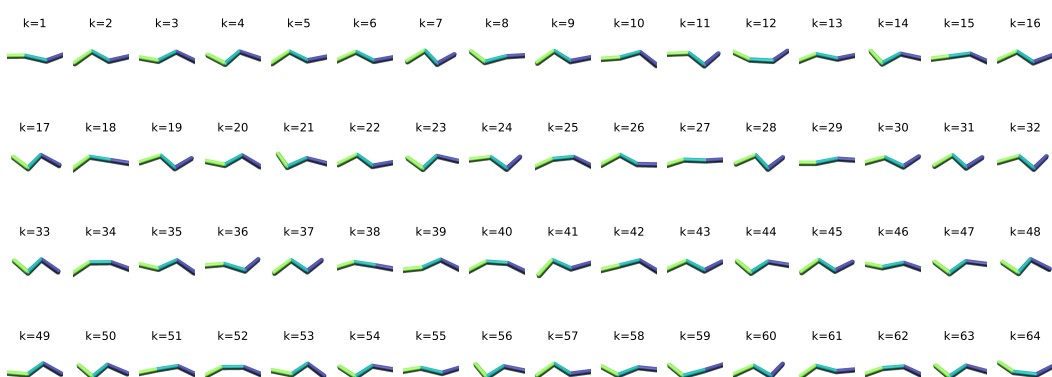

Figure 10: MuJoCo Swimmer-v5 poses resulting from continuously applying the mode of each learned component of a 64-component Gaussian mixture policy as the action. These results illustrate the diversity and coverage of the continuous action space.

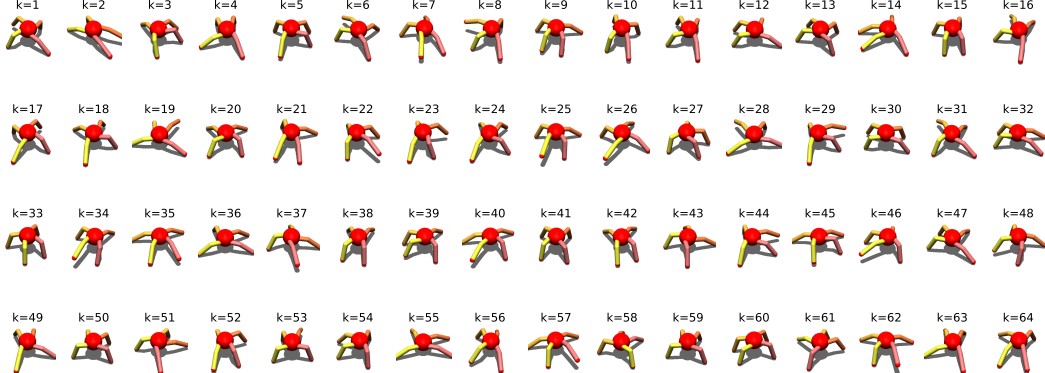

Figure 11: MuJoCo Ant-v5 poses resulting from continuously applying the mode of each learned component of a 64-component Gaussian mixture policy as the action, showing diversity and coverage of the continuous action space.

Table 7: Hyperparameters

| Parameter | Value |
|---|---|
| Optimizer | Adam (Kingma & Ba, 2014) |
| Learning rate | $3 \times 10^{-4}$ |
| Discount $(\gamma)$ | 0.9 |
| Replay buffer size | $10^5$ |
| Number of hidden layers (all networks) | 2 |
| Number of hidden units per layer | 256 |
| Number of samples per minibatch | 32 |
| Steps per epoch | 2000 |
| Initial random steps | 2000 |
| Nonlinearity | ReLU |
| Target network smoothing coefficient $(\tau)$ | 0.005 (Polyak & Juditsky, 1992) |

---

**Algorithm 1** Discrete Policy Updates

---

**Require:** A finite state space $\mathcal{S}$, finite action space $\mathcal{A}(s)$, transition kernel $p(s'|s,a)$, discount factor $\gamma \in [0,1)$, specialization parameter $0 \leq \alpha < 1$, number of components $K(s)$, number of iterations $N$. (Sums over $a$ and $k$ are over ranges $0, \cdots, |\mathcal{A}|(s)$ and $1, \cdots, K(s)$).

**Ensure:** Sequence $\{V^{(n)}, \pi^{(n)}, w^{(n)}, r^{(n)}\}_{n=0}^{N}$ converging to a fixed point.

1: **Initialize:**

2: $\quad V^{(0)}(s) \leftarrow 0, \quad \forall s \in \mathcal{S}$

3: $\quad$ Choose arbitrary $\pi_k^{(0)}(a|s) > 0, \ w_k^{(0)}(s) > 0, \ \forall k, s, a$

4: $\quad$ Compute initial mixture policy $\pi_m^{(0)}(a|s) \ = \ \sum_k w_k^{(0)}(s) \, \pi_k^{(0)}(a|s)$

5: $\quad$ Compute initial responsibilities $r_k^{(0)}(s,a) \ \leftarrow \ \frac{w_k^{(0)}(s)\,\pi_k^{(0)}(a|s)}{\pi_m^{(0)}(a|s)}, \quad \forall k, s, a$

6: **for** $n = 1$ to $N$ **do**

7:

8: $\quad$ **for all** $s \in \mathcal{S}$ **do**

9: $\quad\quad$ Compute $Q^{(n-1)}(s,a) \ \leftarrow \ \sum_{s'} p(s'|s,a)\, V^{(n-1)}(s'), \ \ \forall a$

10: $\quad\quad G(s) \ \leftarrow \ G\big(s;\, \pi^{(n-1)}, w^{(n-1)}, r^{(n-1)}\big)$

11: $\quad\quad V^{(n)}(s) \ \leftarrow \ G(s) \ + \ \gamma \sum_a \pi_m^{(n-1)}(a|s)\, Q^{(n-1)}(s,a)$

12: $\quad$ **end for**

13:

14: $\quad$ **for all** $k = 1, \ldots, K, \ s \in \mathcal{S}, \ a \in \mathcal{A}$ **do**

15: $\quad\quad \pi_k^{(n)}(a|s) \ \leftarrow \ \dfrac{\big[r_k^{(n-1)}(s,a)\big]^{1/(1-\alpha)} \exp\big(\frac{\gamma}{1-\alpha}\, Q^{(n-1)}(s,a)\big)}{\sum_{a'} \big[r_k^{(n-1)}(s,a')\big]^{1/(1-\alpha)} \exp\big(\frac{\gamma}{1-\alpha}\, Q^{(n-1)}(s,a')\big)}$

16: $\quad$ **end for**

17:

18: $\quad$ **for all** $s \in \mathcal{S}, \ k = 1, \ldots, K$ **do**

19: $\quad\quad w_k^{(n)}(s) \ \leftarrow \ \dfrac{\big(\sum_a [r_k^{(n-1)}(s,a)]^{1/(1-\alpha)} \exp \frac{\gamma}{1-\alpha} Q^{(n-1)}(s,a)\big)^{1-\alpha}}{\sum_{l=1}^{K} \big(\sum_a [r_l^{(n-1)}(s,a)]^{1/(1-\alpha)} \exp \frac{\gamma}{1-\alpha} Q^{(n-1)}(s,a)\big)^{1-\alpha}}$

20: $\quad$ **end for**

21:

22: $\quad$ **for all** $s \in \mathcal{S}, \ a \in \mathcal{A}, \ k = 1, \ldots, K$ **do**

23: $\quad\quad$ Compute new mixture policy $\pi_m^{(n)}(a|s) \ = \ \sum_k w_k^{(n)}(s)\, \pi_k^{(n)}(a|s)$

24: $\quad\quad$ Compute new responsibilities $r_k^{(n)}(s,a) \ \leftarrow \ \frac{w_k^{(n)}(s)\,\pi_k^{(n)}(a|s)}{\pi_m^{(n)}(a|s)}$

25: $\quad$ **end for**

26: **end for**

---

---

**Algorithm 2** Learning Components using maxi-mix-mini-com framework for continuous domains

---

1: **Input:** Policy parameters $\theta$, Q-function parameters $\phi_1, \phi_2$, target Q-function parameters $\phi_1^{\text{targ}}, \phi_2^{\text{targ}}$, specialization parameter $\alpha$, discount factor $\gamma$, target update rate $\rho$, learning rate $\eta$, Gumbel–Softmax temperature $\tau$
2: Initialize replay buffer $\mathcal{D}$
3: **for** each iteration **do**
4:     **for** each environment step **do**
5:         Sample component index: $k_t \sim \text{GumbelSoftmax}\big(w_\theta(s_t), \tau\big)$
6:         Sample action: $a_t \sim \pi_{k_t,\theta}(\cdot | s_t)$
7:         Execute $a_t$ in the environment; observe next state $s_{t+1}$, and terminal flag $d_t$
8:         Store transition $(s_t, a_t, s_{t+1}, d_t)$ in $\mathcal{D}$
9:     **end for**
10:     **for** each gradient step **do**
11:         Sample mini-batch of transitions $\{(s_i, a_i, s'_i, d_i)\}_{i=1}^B \sim \mathcal{D}$
12:         **Update Q-functions:**
13:         **for** each sample $i$ in the mini-batch **do**
14:             Sample component for next state: $k'_t \sim \text{GumbelSoftmax}\big(w_\theta(s'_i), \tau\big)$
15:             Sample next action: $a'_i \sim \pi^b_{k'_i,\theta}(\cdot \mid s'_i)$
16:             Compute target:

$$y_i = \gamma(1 - d_i)\Big[Q_\phi^{\text{targ}}(s'_i, a'_i) - \log \pi_{m,\theta}(a'_i|s'_i) + \alpha \log \pi^b_{k'_i,\theta}(a'_i|s'_i)\Big]$$

17:         **end for**
18:         Update Q-functions by minimizing:

$$\phi_j \leftarrow \arg\min_{\phi_j} \frac{1}{B} \sum_{i=1}^B \Big(Q_{\phi_j}(s_i, a_i) - y_i\Big)^2, \quad j \in \{1, 2\}$$

19:     **Update Policy:**
20:     For each state $s_i$, sample a component and action using the current policy:

$$k_i \sim \text{GumbelSoftmax}\big(w_\theta(s_i), \tau\big), \quad \tilde{a}_i \sim \pi_{k_i,\theta}(\cdot \mid s_i)$$

21:     Compute the policy gradient:

$$\nabla_\theta J(\pi_\theta^m) \approx \frac{1}{B} \sum_{i=1}^B \nabla_\theta \Big[Q_\phi(s_i, \tilde{a}_i) - \log \pi_{m,\theta}(\tilde{a}_i \mid s_i) + \alpha \log \pi_{k_i,\theta}(\tilde{a}_i|s_i)\Big]$$

22:     Update policy parameters: $\theta \leftarrow \theta + \eta \, \nabla_\theta J(\pi_\theta^m)$
23:     **Update Target Networks:** $\phi_j^{\text{targ}} \leftarrow \rho \, \phi_j^{\text{targ}} + (1 - \rho) \, \phi_j, \quad j \in \{1, 2\}$
24:     **end for**
25: **end for**

---

