# OpenReview forum: "Unsupervised Behavioral Tokenization and Action Quantization via Maximum Entropy Mixture Policies with Minimum Entropy Components"
_ICLR.cc/2026/Conference — Submitted to ICLR 2026_

### Official Review · Reviewer_JUW5 · 2025-10-31

**Soundness:** 3
**Presentation:** 3
**Contribution:** 2
**Rating:** 4
**Confidence:** 3

**Summary:**

The authors present an unsupervised method based on online RL to learn efficient tokenisation of actions in a continuous control problem, by learning a mixture policy that maximises the entropy of the actions while the sub-policies in the mixture minimise the entropy. The authors then use their method to derive quantised actions that can be then optimised through an actor-critic algorithm to solve control problems more efficiently, and they show how the same action quantisation generalises to different tasks. They provide empirical results showing how the method is comparable or improves over existing work.

**Strengths:**

- The problem of how to compress complex RL problems into interpretable, efficient sub policies or skills is very relevant.
- The solution proposed by the authors (the maxi-mix-mini-com objective) is interesting and I think unsupervised methods are a reasonable approach to solve this problem.
- The empirical evaluation and the baselines compared are sufficient to demonstrate the efficacy of the method.

**Weaknesses:**

- I am not completely convinced by the method as is, see the box below for questions.
- The method seems to rely on quite a few heuristics, which are not always clearly explained. For example, the authors state in the limitations that they need to downscale the variance of the learned component policies a posteriori. It is not very clear how much engineering the method needs to work at scale.

**Questions:**

I write the questions in order of appearance in the text (not in order of relevance).

1. What does long-lived or short-lived mean, intuitively and formally?
2. Would the objective in (2) be maximised by a uniform mixture of deterministic policies?
3. If this is the case (and please correct me if i'm wrong), I’m not sure why this requires learning. Couldn't you construct these policies online with zero learning cost? At each step, you construct the sub-policies by as deterministic (or very low variance) actions, sequentially such that each new policy is sufficiently different from the already generated ones, and then you randomise with equal probability for the mixture.
4. Continuing on this line, what is preventing from converging to trivial (useless) policies? How can you assure that the sub-policies learned are of any interest? A trivial solution to the objective (if I understand correctly), with e.g. 2 policies, would be each policy picks one action deterministically, and both policies are mixed with equal weights, but this is not necessarily interesting in many cases.
5. Shouldn't the unsupervised ‘tokenisation’ be somehow linked to other policy metrics like obtained rewards? In line 193 you mention that the method leads to a compression of the actions, but can this lead to an ‘arbitrary’ compression that does not take into account rewards (and in fact it will not take them into account). So is it possible that the quantisation learned completely stops the agent from being able to get high rewards? I can think of toy examples where this would happen and i’m not convinced this does not happen in general. If this is correct, then the fact that the quantised policies work in the examples you tested is because the reward function is expressive enough, perhaps. Please correct me if I misunderstood some part.
6. What is the intuition for the method doing much better than PPO? I would expect an unconstrained algorithm (in terms of the sub-policies) to still do better, but at a higher complexity cost.

---

> ### Author Response · Authors · 2025-11-20
> **Author Response (part 1 of 4)**
>
> Dear Reviewer,
>
> Thank you for your thoughtful review and for your positive assessment of the problem, the maxi–mix–mini–com objective, and the empirical evaluation. In the point-by-point responses below, we address your concerns.
>
> &nbsp;
>
> >Response to Q1: Long-lived vs. short-lived components
>
> Thank you for asking us to clarify this; the distinction between “long-lived” and “short-lived” components is indeed central to our formulation, and has been clarified in the Introduction, Lines 063-066.
>
> Intuitively, by long-lived we mean the standard notion of options or skills in hierarchical reinforcement learning: a discrete latent variable (option or skill) is selected and then executed for many environment steps before termination, implementing a temporally extended behavior such as “run to the next room” or “move around the obstacle”. By contrast, short-lived components are discrete behavioral choices that are intended to last for only one (or very few) time steps. In our work we focus on the extreme case: each token is a one-step behavioral primitive that can be arbitrarily recombined from step to step, so that complex behaviors emerge purely from their composition rather than from long internal rollouts of a single skill.
>
> Formally, long-lived components correspond to the usual options/skill-discovery setting, where a discrete latent $z$ (or option $o$) is sampled at some time $t_0$ and then held fixed for $H \gg 1$ steps. A typical parameterization is
>
> $$
> a_t \sim \pi(a_t \mid s_t, z), \qquad t_0 \le t < t_0 + H,
> $$
>
> together with a separate termination rule that determines when a new (z) is chosen. The lifetime of a component is the expected number of consecutive steps for which the same (z) remains active, which is strictly greater than one for long-lived options or skills.
>
> In our setting, the components ${\pi_k(a \mid s)}_{k=1}^K$ have the same conditional form as in options or skills, but are used in a strictly short-lived way: the discrete index $k_t$ is (in principle) resampled at every time step from the mixture weights (or a discrete controller) $w(\cdot \mid s_t)$, and the action is drawn from the corresponding component,
>
> $$
> k_t \sim w(\cdot \mid s_t), \qquad a_t \sim \pi_{k_t}(a \mid s_t).
> $$
>
> Thus, the effective lifetime of a token is one step, and any token can be followed by any other token at the next step. This is what we mean by short-lived behavioral tokens: they share the same mathematical form $\pi_k(a \mid s)$ as options or skills, but are explicitly optimized and used under the assumption that they can be switched at every step, so that the agent can compose arbitrary behaviors by sequencing tokens rather than by committing to a long-horizon skill.
>
> >Response to Q2: Uniform deterministic mixtures
>
> Thank you for the comment and for reading the theoretical results carefully. We apologize if this important point was not sufficiently clear in the submission. The objective in (2) is not generally maximized by a uniform mixture of deterministic policies. In particular, in any environment that has at least one of the following properties: (1) state-dependent action sets (as in our grid world, where wall states admit fewer actions than interior states), (2) terminal states (e.g., zero-energy states in the grid world or falls for Humanoid), or (3) nonlinear dynamics (which holds for essentially all standard continuous-control benchmarks such as Humanoid, Ant, Swimmer, etc.), a uniform mixture of deterministic components is strictly suboptimal under the maxi–mix–mini–com objective. The key result that the solution cannot collapse to a uniform mixture, nor to a mixture with only a single active component, is stated and proved in lines 1068-1079. Theorem 2.
>
> In the tabular setting of Theorem 2, our policy-iteration scheme monotonically increases the value of Eq. (2). If we initialize with non-identical components and strictly positive weights $w_k(s) > 0$ for all $k,s$, the initial value is already higher than that of any collapsed solution, and monotonic improvement implies that the algorithm will not converge to a trivial uniform (or single-component) mixture in this regime. We have clarified this in the Discussion Line 522. and in Sec. 2.1, Lines 293-299.

---

> > ### Author Response · Authors · 2025-11-20
> > **Author Response (part 2 of 4)**
> >
> > &nbsp;
> >
> > >Response to Q3: Necessity of learning vs. handcrafted mixtures
> >
> > Thank you for the follow-up question and for spelling out this alternative construction.
> > First, as clarified above, the maxi–mix–mini–com objective in Eq. (2) is not maximized by a uniform mixture of deterministic components, except in degenerate cases with perfectly symmetric dynamics and no terminal states. In the presence of state-dependent constraints, terminal states, or nonlinear dynamics (which cover all our tabular and MuJoCo settings), a hand-crafted uniform mixture of deterministic policies is strictly suboptimal for the objective in Eq. (2), so the target we are optimizing for cannot be recovered by a simple analytic construction.
> > Second, even if one fixes the idea of ‘a mixture of deterministic policies’’, constructing a useful set of such policies without learning already assumes knowing which actions lead to long-horizon high entropy and which ones quickly terminate or collapse the trajectory. The proposal of ‘’sequentially generating deterministic (or low-variance) actions that are sufficiently different and then randomizing uniformly’’ is essentially a geometric construction in action space (e.g., spreading actions out in Euclidean norm). Our objective, however, is defined in terms of trajectory-level entropy under the true dynamics. Two actions that are far apart in torque space can induce almost identical next-state distributions, while two actions that are close in torque space can have very different long-term effects once dynamics, contacts, and termination are taken into account. Discovering which components are genuinely distinct and useful from the perspective of entropy generation in Eq.  (2) therefore requires interaction with the environment; there is no dynamics-agnostic geometric rule that guarantees high long-horizon entropy.
> > This distinction is most evident on complex systems such as the Humanoid. A purely hand-crafted, ‘’zero-learning’’ construction of deterministic components, for example, uniform or heuristic quantization in action space  is exactly what we implement as our random quantization baselines. As shown in Tables 5–6 and videos 4, such state-independent constructions produce components that very often drive the agent into catastrophic terminal states (falls) and yield extremely low returns. In contrast, the components learned by optimizing Eq. (2) systematically avoid these terminal configurations while still spanning diverse local behaviors, and they support substantially better downstream performance. This empirical gap indicates that, in practice, naïve online construction of deterministic sub-policies without learning is insufficient for high-dimensional, fragile systems, and that learning is needed to align the components with the actual dynamics and termination structure of the environment.
> >
> > >Response to Q4: Avoiding trivial or useless policies
> >
> > Thank you for this question; it goes to the core of why the learned components are non-trivial. When we restrict the agent to a small discrete action set, a useful bank of components should (i) avoid driving the agent into terminal or low-action states, (ii) provide different local ways of acting from the same state, and (iii) be sharp enough that a high-level controller can reliably select each component for a specific behavior. The maxi–mix–mini–com objective in Eq. (2) is constructed so that policies violating any of these criteria receive low intrinsic return: components that cause frequent termination or collapse into low-dimensional regions sharply reduce future mixture entropy; components that behave identically at a state fail to increase mixture entropy; and diffuse, noisy components are directly penalized through their own entropy term.
> > In the tabular setting, Theorem 2 shows that our policy-iteration scheme monotonically improves Eq. (2) and does not admit collapsed, redundant collections of components as fixed points when initialized with non-identical components and positive weights. Empirically, we observe the same effect in continuous control: random or heuristic quantization, which does not enforce these three criteria, leads to frequent falls and poor downstream performance, whereas components learned under maxi–mix–mini–com avoid terminal states, remain non-collapsing, and provide distinct, specialized behaviors that a discrete controller can effectively sequence.

---

> ### Author Response · Authors · 2025-11-20
> **Author Response (part 3 of 4)**
>
> &nbsp;
>
> > Response to Q5: Ensuring sub-policies are meaningful
>
> Thank you for this question; it highlights an important potential failure mode of our objective and gives us a chance to clarify it.
>
> The trivial construction you describe—two sub-policies that each pick a single fixed action deterministically and are mixed with equal weights—is both state-independent and time-local. It does not maximize our objective in (2) once environment dynamics, terminal states, and action bounds are taken into account. Recall that (2) combines a long-horizon entropy term for the mixture policy with a penalty on component entropies:
>
> $$
> R(s; \pi, w) ;=; H(\pi_m(\cdot \mid s_t)) - \alpha \sum_k w_k(s_t),H(\pi_k(\cdot \mid s_t)),
> $$
>
> and this intrinsic reward is accumulated along trajectories under the true dynamics. In any environment with (i) terminal states, (ii) state-dependent action availability, or (iii) nonlinear dynamics (all of our grid-world and MuJoCo benchmarks satisfy at least one of these), a state-independent uniform mixture of constant deterministic actions will typically either (a) drive the agent into terminal or low-entropy regions, where future entropy collapses to zero, or (b) spend most of its time in regions where only a strict subset of actions is effective. In both cases, there exists a state-dependent mixture that maintains higher long-horizon $H(\pi_m)$ while keeping components low-entropy, so the trivial uniform construction is strictly suboptimal for (2). In the tabular setting, this is made precise by our analysis: collapsed or uniform mixtures are not local optima of (2), and the policy-iteration scheme in Theorem 2 monotonically increases the objective and does not converge to such solutions under mild initialization assumptions.
>
> Crucially, in our method both the component policies $\pi_k(a \mid s)$ and the mixture weights $w_k(s)$ are functions of the state. Components may become nearly deterministic at a given state (minimizing their own entropy), but they implement different behaviors in different regions of the state space, and the mixture weights adapt so as to avoid terminal states and low-entropy bottlenecks while spanning diverse locally available behaviors. This state dependence is what makes the learned sub-policies “of interest”: on agents such as Ant or Humanoid, the same bank of components yields distinct movement modes in safe upright states and distinct stabilizing responses near falls, which both preserve future entropy and support high downstream returns. By contrast, state-independent deterministic components with uniform weights, or random / hand-designed quantization sets, perform poorly under our objective and in downstream control tasks (as shown by our random AQ and USD baselines). Taken together, the theoretical properties of (2) and the empirical comparisons indicate that the learned sub-policies are non-trivial, state-dependent behavioral tokens rather than uniformly mixed, uninformative deterministic actions.

---

> > ### Comment · Reviewer_JUW5 · 2025-11-25
> >
> > I thank the authors for the detailed rebuttal. In particular this comment I found very useful and insightful. I think the corresponding extract added in the discussion is quite valuable.
> >
> > I don't have further questions and I'll increase my score accordingly.

---

> > > ### Author Response · Authors · 2025-11-25
> > >
> > > Thanks a lot for your quick and positive feedback. We highly appreciate all your previous comments, and we are happy to see that they have been satisfactorily addressed in our responses.

---

> ### Author Response · Authors · 2025-11-20
> **Author Response (part 4 of 4)**
>
> &nbsp;
>
> >Response to Q6: Relation between unsupervised tokenization and rewards
>
> Thank you for this question; it raises an important limitation of any purely unsupervised tokenization scheme. In our work, the unsupervised stage is explicitly intended to learn task-agnostic behavioral tokens, not to optimize any particular reward. The maxi–mix–mini–com objective is not an arbitrary compression in action space: it is shaped by the dynamics and termination structure. By maximizing long-horizon mixture entropy while penalizing component entropy, it encourages a bank of components that (i) avoid terminal and low-action regions where future entropy collapses, and (ii) span distinct local ways of acting at each state with sharp, specialized behaviors. In that sense, the compression is constrained by what the system can do safely and richly under its dynamics, rather than by a purely geometric or random criterion.
>
> At the same time, we fully agree that, in general, there is no guarantee that such an unsupervised compression preserves optimal performance for every possible reward function. One can construct toy MDPs where high reward depends on very narrow, low-occupancy behaviors that are disfavoured by a maximum-occupancy objective, and with a very small number of tokens those behaviors might indeed be excluded. We therefore do not claim universal optimality of the quantized action set. What our results show is that, on standard continuous-control benchmarks such as Ant and Humanoid, the structural bias of our objective (stay alive, remain in high-action regions, generate rich local behaviors) is well aligned with the typical reward design (forward progress, uprightness, moderate control cost). In these domains, a bank of safe, span-seeking locomotion and stabilization token is sufficient to achieve returns close to PPO and clearly better than random quantization and unsupervised skill-discovery baselines, indicating that the compression is not arbitrary in practice.
>
> Finally, there is a natural way to handle settings where unsupervised tokens and reward might be misaligned. In the tabular analysis, with sufficiently many components the mixture can represent any maximum-entropy policy, so any loss of optimality is fundamentally a capacity choice (how small we choose the number of tokens) rather than a flaw of the objective itself. More generally, one can combine extrinsic reward with the maxi–mix–mini–com term during downstream training, using our objective as an intrinsic regularizer while allowing controlled fine-tuning of the components. In this paper we deliberately keep the tokens frozen to isolate the value of a purely unsupervised prior. We thank the reviewer for bringing all these important concerns.
>
>
> >Response to Q7: Why tokens can outperform PPO
>
> Thanks again for the question. Indeed, we were initially puzzled by this result as well. Our intuition for why our method can match or sometimes outperform continuous PPO under this protocol is that the learned token bank provides a strong, dynamics-aware inductive bias that simplifies the downstream optimization problem. Continuous PPO in the original action space must simultaneously discover safe locomotion, avoid terminal states, and optimize the task reward from scratch in a high-dimensional action space. In contrast, our unsupervised stage first learns a bank of state-dependent, safe, span-seeking behavioral tokens; the downstream PPO then only needs to learn how to sequence these tokens in a discrete action space. In other words, PPO with our tokens operates on a constrained, well-structured action manifold already aligned with the system’s dynamics, which can make optimization substantially easier with fixed hyperparameters and finite data. This can lead to realized performance that is comparable to or better than a continuous PPO baseline that has more expressive capacity in principle but a more difficult optimization problem in practice.
>
> &nbsp;
>
> We thank you again for your detailed questions and comments, which prompted us to clarify the intuition behind the objective, explain why the learned components are non-trivial and meaningful, discuss when unsupervised tokenization may or may not preserve reward-optimality, and better document implementation choices such as variance downscaling. We have updated the PDF accordingly, and all modifications are now colored to make them easy to locate.

---

### Official Review · Reviewer_15JY · 2025-10-31

**Soundness:** 2
**Presentation:** 3
**Contribution:** 2
**Rating:** 4
**Confidence:** 3

**Summary:**

The paper proposes an unsupervised method to quantize the action space to learn ‘tokens’, specialized one-step policies, which are used by downstream policies to tackle a wide range of tasks. The key idea behind the proposal is the maxi–mix–mini–com entropy objective: maximize the entropy of the mixture policy to ensure coverage, while minimizing the entropy of each component for specialization. A tractable iterative algorithm is proposed and theoretically shown to converge to a unique solution in the tabular setting. Empirical results indicate that the learned tokens are task-agnostic and can be used to achieve performance comparable to methods explicitly designed for single-task optimization.

**Strengths:**

The paper is well-motivated and clearly structured for the most part. The experiments in the tabular domain effectively illustrate the core idea, while those in continuous control environments demonstrate the scalability of the proposed approach. Furthermore, the supplementary videos help convey the qualitative behavior of the learned representations. The paper presents four theorems that substantiate its main claims and experimental findings. All four proofs appear to be sound. I was able to follow and did not identify any errors in the proofs of Theorems 1 and 2. The proofs of Theorems 3 and 4 are comparatively straightforward, as they build directly upon the results established in [1] and [2].

**Weaknesses:**

However, the paper has three issues:

An important advantage of unsupervised behavioral tokenization is its potential to improve sample efficiency in downstream tasks. As stated in Lines 42–43, “By focusing on core representative tokens, behavioral tokenization can improve sample efficiency, accelerate convergence, and avoid wasteful exploration of irrelevant continuous actions”. However, the paper reports only the final downstream performance after 3 million training steps, without intermediate evaluations. Presenting learning curves or periodic evaluations for the proposed method and the baselines would provide clearer empirical support for this claim.

The paper does not specify how the hyperparameters were chosen for both the proposed method and the baselines. Were defaults used? If any hyperparameters were tuned, did each method get a fair opportunity to tune the same number of hyperparameters? Were hyperparameters tuned across environments?

The reported results are based on only five random seeds. Strong claims cannot be made based on such a small number. You could either increase the number of seeds, and then justify why it is sufficient, or you could aggregate across environments and only make claims at the aggregate level. You could then just show individual runs per environment, to qualitatively show behavior.

Finally, though not strictly a weakness, it was unclear why the modes of the component Gaussian policies were used as discrete actions. To quote:
“Action space is quantized using the modes of the learned component Gaussians, a method that provides the first unsupervised online AQ algorithm. Next, state-dependent quantized actions are used as discrete actions in a DQN (Mnih et al., 2013) or discrete PPO (Schulman et al., 2017). The learned components and quantization are not allowed to change during the optimization of cumulative reward using the discrete controller”
Couldn’t the component policies themselves be considered the tokens, and DQN learn to take the action of sampling from one of the component policies? This is not obviously better, but it would be useful to discuss this choice more.

(Putting Minor Points here, as there is no separate box. These are not major issues)
1. It was not immediately clear what the state- or action-centric perspective referred to. The concept only became clear after reading Section A.1 (Lines 799–809), which appears too late in the paper. I would suggest moving parts of this explanation to the introduction for better clarity and accessibility.

2. The term d_{\pi_{m,\theta}} is introduced in Theorem 3 but defined only later in Theorem 4.

3. Line 329: citation formatting should be corrected to “from Jang et al., 2016” (without brackets).

4. The references section requires proper formatting.

5. Line 1309 – It appears that M in \mathbb{R}^M has not been defined earlier. Based on the context, it seems to correspond to | \mathcal{A} |.

6. In Section A.5, the paragraph on “Downstream Tasks” is missing the word “Table” when referring to the results.

References:
[1] Sutton, Richard S., and Andrew G. Barto. "Reinforcement learning: an introduction, 2nd edn. Adaptive computation and machine learning." (2018)
[2] He, Jiamin, et al. "Investigating Mixture Policies in Entropy-Regularized Actor-Critic."

**Questions:**

Questions, summarized from the above weaknesses.

1. Could you provide results on learning efficiency?

2. Could you explain how hyperparameters were chosen?

3. Could you justify claims based on the current number of seeds, or provide updated results to support claims?

4. Can you explain the choice of using modes of the Gaussian component policies?

I have currently put scores based on uncertainty on these points. For example, I cannot assess soundness without understanding how hyperparameters were chosen, so even though the theory is sound, I had to put a lower rating there. I will adjust this based on responses to questions.

---

> ### Author Response · Authors · 2025-11-20
> **Author Response (part 1 of 2)**
>
> Dear Reviewer,
>
> Thank you for your careful and constructive review of our submission, and for your positive comments on the motivation, structure, experiments, and theoretical results. In the point-by-point responses below, we address your concerns.
>
> &nbsp;
>
> >Response to Q1: Learning efficiency evidence
>
> We sincerely apologize for not providing this important analysis. We now include full training curves for Humanoid, Ant, Hopper, Swimmer, BipedalWalker, Keepbalance, and LunarLander (see new Fig. 7). These plots show, for example, that a Humanoid agent initialized with pretrained components starts around return 4000 because it has already learned not to fall, whereas continuous SAC/PPO must learn this from scratch. Across the other systems the discrete controller operating on our fixed token bank closely tracks the continuous baselines throughout training. These results substantiate the sample-efficiency improvements that were previously only described qualitatively.
>
> >Response to Q2: Hyperparameter selection protocol
>
> Thank you for raising this point; we agree it should be stated explicitly, and we have improved the description in Sec. A.6, line 1527 of the revised manuscript.
>
> For our method, all hyperparameters apart from the specialization parameter alpha and the number of components K are fixed across environments and reported in Appendix.C (Table. 7). We do not tune these values per environment. The only method-specific degrees of freedom are alpha and K. Instead of selecting them by task-specific tuning, we evaluate a small grid of $(\alpha, K)$ combinations and report the resulting performance and diversity statistics (e.g., Table~6). The main results use representative $(\alpha, K)$ settings from this grid and are not optimized separately for each benchmark.
>
> For the downstream RL algorithms (DQN, discrete PPO, SAC, and continuous PPO), we use the standard default hyperparameters from widely used reference implementations (stable baselines), with identical network architectures and training budgets across all methods on a given environment. These hyperparameters are not tuned for our method or for any baseline. For the unsupervised skill-discovery baselines, we use the official implementation of METRA released by Park et al. (2024), including their recommended hyperparameters, without additional tuning.
>
> Consequently, no method receives environment-specific hyperparameter optimization. In particular, discrete PPO and DQN are run with the same configuration when trained on our components, on random quantized action sets, and on USD skills, so that performance differences can be attributed to the underlying primitive/action set rather than to differences in high-level RL tuning.
>
> >Response to Q3: Number of random seeds
>
> Thank you for this remark; we agree that the number of seeds and the breadth of environments should be made explicit. In the revised experiments, we have increased the number of random seeds from 5 to 10 for all methods and added three additional environments to our benchmark suite. All results in the new figures and tables are therefore based on 10 independent runs per method and environment, providing a more robust estimate of performance and variability. No important changes are observed, and therefore all our main statements remain. Note also that we now provide confidence intervals, instead of standard errors, in Tables 1-3 to make clearer the significant differences between methods, or lack thereof.

---

> ### Author Response · Authors · 2025-11-20
> **Author Response (part 2 of 2)**
>
> &nbsp;
>
> >Response to Q4: Using component modes as discrete actions
>
> Thank you for highlighting the lack of clarity in this design choice. Conceptually, we do treat the component policies ${\pi_k(\cdot \mid s)}$ themselves as the behavioral tokens; the discrete controller always selects a token index $k$. As noted in Fig. 1(c) and Sec. 4, there are two natural ways to ground this choice into a continuous control signal: either (i) execute a sampled action $a_t \sim \pi_k(\cdot \mid s_t)$, or (ii) execute a deterministic representative, e.g., the mode $\mu_k(s_t)$.
>
> In the experiments reported in the main text we use the modes $\mu_k(s)$ as state-dependent discrete actions for three practical reasons. First, this gives a one-to-one mapping $(s,k) \mapsto a$ and therefore induces a standard discrete-action MDP, which allows us to plug our tokens directly into off-the-shelf DQN and discrete PPO and to compare fairly to existing action-quantization baselines that are defined in terms of deterministic prototype actions. Second, using the modes removes additional stochasticity between the discrete controller and the environment, reducing variance in value estimation and isolating the effect of the learned token bank rather than conflating it with sampling noise. Third, our unsupervised objective explicitly drives each component $\pi_k(\cdot \mid s)$ toward low entropy and strong specialization; in this regime the mode $\mu_k(s)$ is a good summary of the component’s behavior, and we observed qualitatively similar behaviors when sampling from $\pi_k$ instead of using $\mu_k$ (see Fig. 1 and Videos 2–3).
>
> That said, nothing in our framework prevents us from using the stochastic component policies directly as tokens—i.e., letting the discrete policy choose $k$ and then sampling $a \sim \pi_k(\cdot \mid s)$ for control. This is a compatible variant that simply leads to a higher-level policy over one-step stochastic skills rather than over deterministic quantized actions.
> We have clarified in Lines. 425 and 1344 of the revision that (i) the behavioral tokens are the component policies $\pi_k$, (ii) our reported results instantiate them via their modes for stability and comparability with standard action quantization.
>
> >Response to M1: State- vs. action-centric perspective
>
> Thanks for noticing this lack of clarity in the introduction. We have now moved the core definitions and concrete examples from Section A.1 to the Introduction section (Lines 070-075).
>
> >Response to M2: Definition of $d_{\pi_{m,\theta}}$
>
> Thank you for pointing out the missing definition. In the revised Theorem.3  (Lines 318) we now state explicitly that $d_{\pi_{m,\theta}}(s)$ denotes the discounted state-occupancy measure induced by the mixture policy $\pi_{m,\theta}$.
>
> >Response to M3–M6: Minor formatting fixes
>
> Thank you for carefully checking these presentation details, and we apologize for the earlier inconsistencies. Line 357 now reads “from Jang et al., 2016” without brackets; the entire references section has been reformatted to the conference style; Line 1476 now states explicitly that $M = |\mathcal{A}|$ when introducing $\mathbb{R}^M$; and the “Downstream Tasks” paragraph in Sec. A.5 now refers to “Table.1” when discussing the reported results. These edits resolve all four minor issues you raised.
>
> &nbsp;
>
>
> We thank you again for your detailed feedback, which prompted us to include learning-efficiency results, clarify the hyperparameter and seeding protocol, justify the use of component modes as discrete actions, and improve several presentation details.  We have updated the PDF accordingly, and all modifications are now colored to make them easy to locate.

---

### Official Review · Reviewer_kzvn · 2025-11-01

**Soundness:** 4
**Presentation:** 3
**Contribution:** 4
**Rating:** 8
**Confidence:** 4

**Summary:**

The paper proposes an algorithm called maxi-mix-mini-com that can learn transferable behaviour tokens in an unsupervised manner. Maxi-mix-mini-com aims to maximise the discounted sum of future entropies while minimising the entropy of each behaviour token. By cleverly introducing a new optimisation variable $r$, the authors presented a provably convergent algorithm that can learn diverse behaviours. The authors also empirically demonstrates that by learning a high-level policy based on the learned behaviour tokens, they can easily obtain a high-performing policy, manifesting the reusability and diversity of the learned behaviour tokens.

**Strengths:**

The paper is well-written and easy to understand. In particular, the introduction section provides a great overview of existing research and effectively articulates the significance of their work, making it one of the best introductions I've recently read. Also, the proposed algorithm is backed by rigorous mathematical theorems and extensive empirical analysis on various benchmarks.

**Weaknesses:**

Although the paper is, in general, easy to follow, there are some parts that need further clarifications.

1. Denoting the right-hand side of (4) by Q is a bit misleading, because technically speaking, it is not a Q function.

2. Please add a cross-reference of Figure 4 to Section 2.1.

3. Lines 315-316 are difficult to understand. How is $\pi_{k,\theta}(a\mid s)$ defined?

4. Figures 3(c) and 3(d) needs more explanation. The action distribution would be different for each state.

**Questions:**

I suppose training upon the learned behaviour tokens would drastically speed up the high-level training process. Could you provide learning curves for the **Transfer of learned behavioral tokens across tasks** experiments?

---

> ### Author Response · Authors · 2025-11-20
>
> &nbsp;
>
> Dear Reviewer,
>
> Thank you for your very positive and thoughtful review of our submission, and in particular for your generous comments on the introduction and overall presentation. In the point-by-point responses below, we address your concerns.
>
> >Response to W1: Equation (4) notation
>
> To avoid confusion, we now clarify right after Eq. 4 (Lines 173) that the term $Q(s,a)$ refers to the expected continuation value defined via the Bellman backup and is different from the conventional action-value function because the immediate reward term $R(s; \pi, w)$ is handled separately.
>
> >Response to W2: Figure 4 cross-reference
>
> Thank you for pointing this out; we now explicitly reference Fig. 4 when introducing the grid-world experiment in Sec. 2.1 (Line 266) so readers can immediately see the setup being described.
>
> >Response to W3: Definition of $\pi_{k,\theta}(a \mid s)$
>
> We now clarify in Theorem 1 (Line 336) that $\pi_{k,\theta}(a \mid s)$ is the Gaussian component parameterized by the neural network weights $\theta$, and we instruct readers to see the theorem statement for the precise definition of the mixture and its gradients.
>
> >Response to W4: Explaining Figures 3(c) and 3(d)
>
> We expanded the caption of Fig. 3 to describe what panels (c)–(d) show: “(c,d) Swimmer component action mode vectors (means $\mu_k(s)$) at a representative random state sampled from a trajectory induced by the mixture policy, for (c) $K=16$ and (d) $K=64$ components.” This clarifies that the action distributions are evaluated at a particular state and illustrates how they vary with K (Line 393).
>
> >Response to Q1: Transfer experiment learning curves
>
> We sincerely apologize for missing this important analysis. We now provide the learning curves for Humanoid, Ant, Hopper, Swimmer, and BipedalWalker as representative cases covering high- to low-dimensional systems (see new Fig. 7 in the revised manuscript).
>
> For the humanoid—the hardest and most high-dimensional task—pretrained behavioral tokens drastically accelerate learning of the running task against both continuous SAC and PPO: the starting return is around 4000, providing a golden start, while continuous SAC and PPO must start from scratch. This is because, with our tokenization pretraining, the agent has learned not to fall, and this is a useful behavioral prior in most realistic scenarios involving a humanoid robot.
>
> For Swimmer, Hopper, and Ant, we observe that even with a limited number of fixed components, the discrete algorithm (PPO) retains a substantial fraction of the performance of the continuous-control baselines, even though learning speeds are diverse. In all these tasks, our algorithm remains competitive with continuous PPO, indicating that the discretization induced by behavioral tokens does not fundamentally limit the attainable control performance in these benchmarks.
>
> These results show that pre-trained tokenization with our algorithm drastically improves learning curves in cases where stabilizing the agent is complex and where the presence of many degrees of freedom complicates control.
>
> We would like to emphasize that sampling efficiency is not the main result of our work. Instead, the fact that, with just $K=64$ components (which in the case of Humanoid corresponds to $\approx 1.28$ actions per dimension, since $64^{1/17} \approx 1.28$), we attain performance on par with continuous SAC and better than continuous PPO is the central result.
>
>
> &nbsp;
>
>
> We are grateful for your constructive feedback and for highlighting both the strengths of the work and the points requiring clarification. The revisions to notation, figures, explanations, and the added learning curves are intended to directly address your comments.  We have updated the PDF accordingly, and all changes relative to the original submission are now highlighted in color to make them easy to track.

---

### Official Review · Reviewer_QfN9 · 2025-11-01

**Soundness:** 2
**Presentation:** 2
**Contribution:** 2
**Rating:** 2
**Confidence:** 4

**Summary:**

This paper proposes an online unsupervised approach for action quantization. This approach has the potential to reduce the complexity of the action space by learning a relatively small but useful action set so that downstream tasks could use this more compact action set to learn more efficiently. In the first step, the agent maximizes the entropy of a mixture policy, while minimizing the entropy of individual components of the mixture, which generates distinct but more focused components. In the second step, a discrete-action algorithm treats the learned and fixed components or their mode as actions and maximizes rewards. In theory, the paper shows that 1) the learned action modes of a mixture policy with enough capacity is lossless (covers the whole action space) in the discrete setting, and 2) the gradient estimators for the unsupervised learning of mixture policies. Empirically, the paper illustrates the impact of the strength entropy regularization and capacity of the mixture, and investigates the performance of this approach in continuous-control tasks.

**Strengths:**

This paper has the following strengths:
1. Learning useful sub-policies in reinforcement learning (RL) is an important problem and has the potential of scaling up RL to solve more complex problems.
2. The proposed unsupervised learning objective for learning diverse mixture components is novel and demonstrated to be effective.
3. It provides useful theoretical characterizations of the proposed method. While the theorems do not appear to be difficult to prove technically, they are useful to describe some of the fundamental properties of learning the mixture policy with the  unsupervised learning objective.
4. The paper is well-written. It’s easy to follow and to find specific details in the appendix.

**Weaknesses:**

Despite the strengths, the paper has some weaknesses to be addressed:
1. The benefit of the proposed approach is not well demonstrated. Since pretraining is performed to extract behavioral prior, the paper does not demonstrate such an approach improves efficiency compared to methods that train from scratch (SAC / PPO).
2. The current empirical evaluation is quite limited. 1) Experiments are only performed in a subset of MuJoCo environments and a few other stand alone tasks. It’d be beneficial to increase the coverage of the tested tasks. For example, more MuJoCo locomotion and navigation environments. 2) The lack of comparison to the natural, trivial uniform quantization.
3. Some limitations of the proposed approach are not discussed: 1) One of the limitations of the supervised learning approach is that it requires a state-dependent action space to encourage meaningful learned components. If such “available action sets” are not available, the learned components will likely just randomly scatter across the action space. 2) Another limitation is that the paper does not investigate how the components could be fine-tuned in down-stream tasks when pretrained components are not optimal and even be limited.

**Questions:**

Here are questions that might impact the rating:
1. The paper mentions the learned components generalize across environment layouts (Line 96). Could the authors clarify if the mentioned result is in Table 1? Further, could the authors clarify the observation spaces in pretraining and downstream training?
2. Could the authors provide learning curves for the experiment results in Section 4?
3. Could the authors provide comparisons to the trivial uniform quantization?
4. Whether DQN or PPO is used for each experiment? It’s unclear from the discussion in Lines 408-409.
5. As mentioned in the appendix, the proof of Theorem 4 appears to be similar to that of a theorem in He et al. (2025). In addition, I also found some parallels between Section 3 and Section 4 of He et al.. Could the authors clarify the difference between the two and highlight the contributions in this paper?

Other minor suggestions:
1. Line 362: It’s confusing to see the acronym SD. Similarly, it might be better to spell out USD in the caption of Table 2.
2. Standard errors are proper confidence intervals. It’s difficult to tell the statistical significance of the reported results in the tables. It’d be better to use proper confidence intervals.

---

> ### Author Response · Authors · 2025-11-20
> **Author Response (part 1 of 3)**
>
> Dear Reviewer,
>
> Thank you for your detailed and constructive review of our submission and for your positive assessment of the proposed method and its presentation. In the point-by-point responses below, we address your concerns.
>
> &nbsp;
>
> >Response to Q1: Layout generalization and observation spaces
>
> Thank you for pointing this out; we appreciate the opportunity to clarify it. The claim that the learned components generalize across environment layouts refers to the Ant results reported in Table 1. In that experiment, the mixture components are pretrained once in the standard, open Ant environment without any maze walls. For the downstream maze-navigation task, we keep this bank of components fixed and only train a discrete high-level controller on top of them in the new layout. As Table 1 shows, the same pretrained components support a variety of maneuvers and directions that allow the agent to reach the goal in the maze, despite never having seen that layout during pretraining.
>
> Regarding observation spaces, pretraining and downstream training share the same proprioceptive state representation $s_t$ of the agent (joint angles, joint velocities, and other body-centric features), and pretraining does not include any information about a specific task or environment layout. In the downstream goal-conditioned tasks, we follow the standard formulation and augment the input to the discrete controller with both the achieved and desired goal, that is, it receives ($s_t$, achieved_goal_t, desired_goal). In contrast, the low-level components always operate on $s_t$ alone and never see the achieved or desired goal. This design emphasizes the intended property of our method: the components are learned as dynamics- and embodiment-specific behavioral tokens, independent of any particular layout or goal specification, while task- and layout-specific information is handled exclusively by the high-level discrete controller.
>
> We have explained these points more explicitly in lines 1503–1510 of the revised manuscript.
>
> &nbsp;
>
> > Response to Q2 and W1: Benefit evidence and learning curves
>
> We sincerely apologize for missing this important analysis. We now provide the learning curves for Humanoid, Ant, Hopper, Swimmer, BipedalWalker, and LunarLander as representative cases covering high- to low-dimensional systems (see new Fig. 7 in the revised manuscript).
>
> For the humanoid—the hardest and most high-dimensional task—pretrained behavioral tokens drastically accelerate learning of the running task against both continuous SAC and PPO: the starting return is around 4000, providing a golden start, while continuous SAC and PPO must start from scratch. This is because, with our tokenization pretraining, the agent has learned not to fall, and this is a useful behavioral prior in most realistic scenarios involving a humanoid robot.
> For Swimmer, Hopper, and Ant, we observe that even with a limited number of fixed components, the discrete algorithm (PPO) retains a substantial fraction of the performance of the continuous-control baselines, even though learning speeds are diverse.
>
> In all these tasks, our algorithm remains competitive with continuous PPO, indicating that the discretization induced by behavioral tokens does not fundamentally limit the attainable control performance in these benchmarks.
>
> These results show that pre-trained tokenization with our algorithm drastically improves learning curves in cases where stabilizing the agent is complex and where the presence of many degrees of freedom complicates control.
>
> We would like to emphasize that sampling efficiency is not the main result of our work. Instead, the fact that, with just $K=64$ components (which in the case of Humanoid corresponds to $\approx 1.28$ actions per dimension, since $64^{1/17} \approx 1.28$, we attain performance on par with continuous SAC and continuous PPO is the central result.

---

> ### Author Response · Authors · 2025-11-20
> **Author Response (part 2 of 3)**
>
> &nbsp;
>
> >Response to Q3 and W2.2: Trivial uniform quantization comparisons
>
> We apologize if this was not clear enough: indeed, our previous manuscript already incorporates comparisons to “natural” trivial quantization schemes, including uniform quantization.
> In particular, Tables 5 and 6 reported baselines using (i) a zero-mean Gaussian quantizer, (ii) a uniform quantizer, and (iii) a per-dimension discrete alphabet {−m,0,m}.
>
> These are intended as simple, hand-designed quantization strategies against which to contrast our learned behavioral tokens.
>
> We note, however, that a quantization method we do not test is binning the action space. The reason for this is that generating a bank of actions based on binning would render the bank too large. For instance, with only three bins per action dimension, the total number of quantized actions would be $3^{17} \approx 1.3 \times 10^8$ in Humanoid, making learning with a discrete controller infeasible. This is six to seven orders of magnitude larger than the $K=16$ or $K=64$ banks that we typically use.
>
> We have improved Lines (480-487) in the new version of the manuscript to make this critical comparison clearer.
>
> Our results in Tables 5–6 in the previous version of our manuscript show that naïve action quantization methods largely underperform our method.
>
> &nbsp;
>
> >Response to Q4: Controller choices per environment
>
> Thank you for pointing this out. We use a discrete PPO controller for all downstream experiments except FetchReach-v2, where we use DQN. Concretely, Ant, Humanoid, Swimmer, Walker2D, AntMaze, Hopper, BipedalWalker, and LunarLander all use discrete PPO as the high-level policy on top of the pretrained components, while FetchReach-v2 uses DQN due to its better empirical performance on this small, low-dimensional discrete action space. This information is already stated in the implementation details (“…the same configuration is used … except that for FetchReach-v2 we use DQN…”), but we agree that it is easy to overlook.
> In the revised version, we have made this explicit in Lines (1514-1516) and in the table captions, Table 3, by stating clearly that PPO is used for all environments except FetchReach-v2, where DQN is used.
>
> &nbsp;
>
> >Response to Q5: Difference from He et al.
>
> Thank you for pointing out a lack of more explicit discussion on the differences between He et al.’s proof and ours. He et al. (2025) work in the standard entropy-regularized reinforcement learning setting: their mixture policy is trained with extrinsic rewards and a soft value function, and their half-reparameterization theorem provides a gradient estimator for this reward-driven objective. In contrast, our work is based on a purely unsupervised objective with no extrinsic reward: we maximize mixture entropy while simultaneously minimizing a weighted sum of component entropies (the maxi-mix–mini-com objective) in order to learn reusable behavioral components—this additional minimization is the most novel aspect. Our Theorem 4 adapts the half-reparameterization technique specifically to this unsupervised objective. We have clarified this in Sec. A.1, Lines (815-822).
>
> &nbsp;
>
> >Response to W2.1: Broader evaluation coverage
>
> Beyond the Ant, Humanoid, Swimmer, Walker2D, FetchReach, and maze-navigation tasks already discussed in the paper, we have now added results on Hopper, BipedalWalker, and LunarLander in the rebuttal (see new Fig. 7).
>
> These additional experiments follow exactly the same pretrain-and-reuse protocol and continuous baselines as in the main text.
>
> The new environments corroborate our previous results (see new Table 3).

---

> ### Author Response · Authors · 2025-11-20
> **Author Response (part 3 of 3)**
>
> &nbsp;
>
> >Response to W3.1: State-dependent action availability
>
> Thanks for this important point. We apologize for not having been clearer in this crucial aspect.
> First, our approach does not critically depend on the availability of state-dependent action sets.
> In the grid-world example in Fig. 4 we indeed have state-dependent action sets (cells next to walls have fewer actions than other cells), but this choice is not central.
>
> We have now regenerated results in the grid world when the cells have exactly the same number of actions, and we have found qualitatively similar results (see new Fig. 5): the obtained maxi-mix–mini–com policies are never uniform. This is because the presence of terminal states is enough to shape the optimal policy: cells near terminal states have vanishing probability of generating actions that make the agent fall into terminal states, and this propagates backward to parent cells in succession.
>
> We have clarified this in Lines (293, 522) in the new version.
>
> We finally note that in continuous domains such as Swimmer and Fetch there are no task-specific terminal states and no handcrafted state-dependent action masks. The action space is fixed across states and is only bounded by standard joint and torque limits. Even in these more “unstructured” settings, our objective does not collapse to a trivial, uniform, state-independent policy: it learns components that are well structured and interpretable (for example, coordinated stretches and contractions in Swimmer).
>
> This happens because the non-linear geometry of the joint-angle space together with action boundaries makes some actions systematically better at generating long-horizon action entropy (for instance, by avoiding saturation at angle limits), and the maxi-mix–mini–com algorithm concentrates components on those dynamically meaningful regions instead of scattering them uniformly.
>
> In conclusion, non-uniform maxi-mix–mini–com policies are obtained whenever (i) there are terminal states and/or (ii) the dynamics are non-linear. Arguably, these are common features in most realistic environments.
>
> >Response to W3.2: Component fine-tuning limitations
>
> Thank you for bringing this important point. Indeed, in all experiments in the paper, the low-level components are never fine-tuned on downstream rewards: once pretrained with the unsupervised maxi–mix–mini–com objective, they are kept fixed, and only the high-level discrete controller is trained for each task.
>
> Our main goal in this work was to study whether a single, task-agnostic bank of components, learned purely from dynamics and embodiment and without further fine-tuning, can serve as a reusable behavioral prior across many tasks on the same system. Allowing joint fine-tuning of the components on downstream rewards might obscure how much performance comes from the unsupervised prior versus task-specific adaptation.
>
> At the same time, the objective we use is compatible with extrinsic rewards. In principle, one could combine task rewards with the maxi–mix–mini–com term and use this combined signal to update the components during downstream training.
> This would allow fine-tuning of the components, improving task performance while still encouraging diverse, specialized behaviors. We did not explore such variants in this paper in order to keep the experimental protocol simple and to isolate the benefits of a frozen, reusable prior.
>
> In the revised version, we have clarified this design choice in Lines 430 and 525 and explicitly stated that designing and evaluating fine-tuning schemes for the components on downstream tasks, using our objective as an intrinsic regularizer, is an important direction for future work.
>
> We sincerely thank the reviewer again for making this important point.
>
> >Response to M1: Clarifying SD/USD
>
> We appreciate the suggestion. We now spell out both SD (in Line 395) and USD (in Line 433) at their first occurrences to avoid confusion for readers.
>
> >Response to M2: Confidence intervals reporting
>
> Thank you for helping us improve the clarity of the results. We now report 95% confidence intervals for all results in Tables 1–3 and in Figure 7.
>
> &nbsp;
>
> We thank you again for your thoughtful comments and suggestions, which helped us improve the experiments, clarify limitations, and better position our contributions. We have updated the PDF accordingly, and all changes relative to the original submission are now highlighted in color to make them easy to track.

---

> > ### Comment · Reviewer_QfN9 · 2025-11-27
> >
> > I appreciate the authors’ detailed responses. My confusions are clarified and my concern regarding the discussion on limitations is addressed.
> >
> > However, I would like to follow up on the following unaddressed main concerns:
> > 1. **Benefit evidence.** It appears that the proposed learned action quantizations do not improve sample efficiency in general. While the authors stated in the rebuttal that the main result is that a small number of pretrained components are sufficient, could you clarify the **practical advantage** of this finding? If it does not yield gains in sample efficiency, does it offer benefits in terms of inference speed, transferability, or robustness? Without a clear performance edge over standard continuous methods, the motivation for using this approach remains unclear.
> > 2. **Trivial uniform quantization comparisons.** Thank you for the explanation regarding the joint binned action space. However, this addresses a different type of discretization than what I intended. Apologize for the confusion, but I was referring to factorized (per-dimension) action discretization, as seen in Tang & Agrawal (2020), which avoids the combinatorial explosion of the joint space. Could the authors compare to such a factorized approach or justify why this standard baseline is not included?
> > 3. **Missing data points for $m=0.5$ and $m=1.0$.** Relevant to the point above, I notice that results for the trivial quantization scheme with $m=0.5$ are missing for Walker2D and Humanoid.
> >   - Could the authors provide these results or explain their absence?
> >   - Crucially, is it possible to also include $m=1.0$? As shown in Seyde et al. (2021), a simple "Bang-Bang" control policy (action space $\{ -1, 0, 1 \}$ per dimension) is often sufficient for efficient learning in continuous control benchmarks. Including this baseline is essential to demonstrate that the proposed method offers value beyond simple boundary-extreme control strategies.
> >
> > Minor points:
> > 1. Learning curves. Thank you for adding the learning curves for some environments. For full transparency, it would be beneficial to include these curves for all tested environments in the final version.
> > 2. Contextualization. Since Theorem 4 builds on the technique in He et al., I suggest the authors update the main text to explicitly contextualize the proposal of Theorem 4 relative to that prior work.
> >
> > Tang, Y., & Agrawal, S. (2020). Discretizing continuous action space for on-policy optimization. AAAI.
> >
> > Seyde, T., Gilitschenski, I., Schwarting, W., Stellato, B., Riedmiller, M., Wulfmeier, M., & Rus, D. (2021). Is bang-bang control all you need? solving continuous control with bernoulli policies. NeurIPS.

---

> > > ### Author Response · Authors · 2025-12-02
> > >
> > > We are happy to see that your concerns about limitations have been addressed. Thanks also a lot for the additional comments. We highly appreciate the opportunity to explain our results more deeply.
> > >
> > > > Benefit evidence
> > >
> > > Thank you for asking for further clarifications on the practical benefits of our approach. The main contribution of the paper is to show that (i) the proposed maxi–mix–mini–com objective produces a behavioral prior that consistently outperforms random / trivial quantization schemes and standard unsupervised skill-discovery methods (DIAYN, DADS, LSD, METRA) for the same discrete capacity budget (Tables 2, 5, 6), and that (ii) a small, task-agnostic bank of learned tokens (typically $K = 16$–$64$) is sufficient to recover a large fraction of SAC/PPO performance on challenging benchmarks such as Ant and Humanoid while acting in a fixed discrete action space whose size does not grow with the action dimension (Table 3). In practice, this provides a compact, **transferable** discrete action interface that can be combined with standard discrete RL or planning, while keeping the low-level dynamics in a shared module that already generalizes across tasks and layouts (e.g., open Ant vs AntMaze).
> > >
> > > Regarding sample efficiency, we agree that in standard MuJoCo tasks, having relatively permissive termination and stable dynamics, pretraining does not always translate into clear efficiency gains over continuous methods trained from scratch. However, even in the standard Humanoid setting we already observe that pretrained tokens provide a strong stabilizing prior and largely enhance **sample efficiency**: the agent quickly learns to avoid falling and starts downstream training from high returns (Fig. 7), whereas SAC/PPO must “rediscover” upright, safe behavior from scratch.
> > >
> > > To show that improved sample efficiency is not exclusive of the Humanoid environments and that indeed **sample efficiency is a general practical advantage of our algorithm in any environment where stabilization is hard** (like in most robotic applications), we designed a variant of Ant-v5 that mirrors the same “do not fall” constraint in a controlled way rather than an engineered, pathological setting – the latter typically used in standard evaluations.
> > >
> > > Concretely, we constrain the torso height to a narrower “healthy band” $z \in [0.6, 0.7]$ compared to the standard default band $[0.3, 1.0]$, the latter being overtly permissive; all episodes start at $z = 0.65$. With the new healthy regime, episodes terminate immediately when this band is left, and given its narrow range it poses a stronger constraint on stabilization.   Importantly, we take the downstream reward to exclusively consist of the forward velocity, excluding any survival bonus, to avoid giving our pretrained tokens an automatic advantage. We first pretrain our components for $500{,}000$ interaction steps using only the maxi–mix–mini–com objective under this termination rule, then freeze them and train a high-level controller for $200{,}000$ steps on the forward-only task. SAC and PPO are trained from scratch for $200{,}000$ steps on the same task. Our method achieves $55.81 \pm 6.62$ return with mean episode length $498.8 \pm 102.67$ steps, whereas SAC reaches $9.41 \pm 3.30$ return with length $8.2 \pm 1.92$, and PPO $5.32 \pm 5.21$ with length $7.4 \pm 2.88$. Therefore, **our algorithm not only achieves an improved sampling efficiency, but it improves performance by more than a factor of 5 over both SAC and PPO in terms of average return under the same training budget**.
> > >
> > > The explanation is that, during pretraining, the only way to achieve high cumulative entropy under the narrow healthy band is to learn components that keep the agent upright for long horizons while spanning diverse actions inside that safe region. The downstream controller thus optimizes over tokens that typically survive hundreds of steps and provide a clean forward-reward signal, whereas SAC/PPO remain in a “fall immediately” regime with very short episodes and little usable signal. Taken together with the Humanoid results, this suggests a general pattern: in environments where stabilization is genuinely hard and failure terminates behavior (a regime common in realistic robotic applications with falls, safety limits, or damage), pretraining **behavioral tokens provides a tangible sample-efficiency and robustness benefit on top of the compact, transferable action representation demonstrated in the previous and new results**. We have added these new results in the Appendix A.6

---

> > > > ### Author Response · Authors · 2025-12-02
> > > >
> > > > >Trivial uniform quantization comparisons
> > > >
> > > > No problem!, and thank you for pointing out Tang & Agrawal (2020); we now better understand the baseline the reviewer had in mind. Their factorized baseline discretizes each action dimension into $K$ bins and learns a task-specific factorized policy $\pi(a\mid s) = \prod_i \pi_i(a_i\mid s)$ that maximizes a specific extrinsic reward. Importantly, the resulting joint policy has support over $K^M$ joint actions (e.g., $15^8 \approx 2.6\times 10^9$ for Ant-v5), whereas in our setting the high-level controller can only choose among $K$ joint components (typically $K\in[16,64]$), posing a more restrictive support. Thus, a capacity-matched comparison in terms of “$K$ available actions” is not well defined: **their scheme does not produce a small bank of $K$ reusable joint tokens**, and extracting such a set would require an additional clustering/compression step that lies outside their framework.
> > > >
> > > > Conceptually, our goal is different: we learn a task-agnostic, unsupervised bank of joint components (sub-policies or quantized actions) that is reused across multiple downstream tasks and layouts, while Tang & Agrawal **train a single task-specific policy and do not evaluate reusability of their discretization on new tasks**. We remind the reviewer that to address the design choice you highlight—per-dimension factorization vs.\ learned joint tokens—we already include a factorized random AQ baseline in which each dimension is discretized independently (actions in ${-m,0,m}$ per dimension; Tables 5–6). This baseline has the same discrete capacity at the high level (the controller still selects among $K$ indices), but assumes no learned joint structure. Across all environments, **our unsupervised joint components significantly outperform this factorized baseline**, indicating that learning state- and dynamics-aware joint behavioral tokens is more effective than per-dimension factorization under the same high-level action budget.
> > > >
> > > >
> > > > >Missing data points for $m=0.5$ and $m=1.0$
> > > >
> > > > We apologize for the missing configurations. In the original submission, we avoided very large magnitudes (m) because in our experiments the agent loses fine control under such large action magnitudes, making the resulting policies limited. We have now extended our evaluation accordingly: for Humanoid, we include (m = 0.4), which is the largest magnitude for which the agent can still operate; for Walker2D, we additionally report (m = 0.5) and (m = 1.0); and for Ant, we report (m = 1.0). These results are now included in Tables 5 and 6, showing poor performance compared to our algorithm, as expected for the reason outlined above.
> > > >
> > > >
> > > > > "Bang-Bang" control policy
> > > >
> > > > Thank you for pointing out the importance of bang–bang–style baselines. We  omitted the largest values of $m$ because (i) near-maximal torques produce visually unnatural motions, and (ii) it is harder to stabilize the most complex environments, like the Humanoid, with large torque values.
> > > > To substantiate our claims, in the revised version we have now added the missing trivial quantization results for Walker2D with $m = 0.5$ and $m = 1.0$, for Humanoid with $m = 0.4$ (the maximum admissible torque magnitude in our Humanoid setup, so $m = 1.0$ is not physically feasible there), and for Ant with $m = 1.0$. These new bang–bang configurations (actions in ${-m,0,m}$ per dimension) are explicitly labeled in the updated Tables 5 and 6, still showing inferior performance compared to our algorithm.
> > > >
> > > > Finally, we note that the method of Seyde et al. (2021), like Tang & Agrawal (2020), discretizes the action space and then trains a single task-specific policy to maximize a specific extrinsic reward; it does not aim to learn an unsupervised, task-agnostic bank of reusable components as in our work. Nevertheless, the two papers are relevant, and we have quoted them in our revised manuscript.
> > > >
> > > >
> > > > >Learning curves
> > > >
> > > >  We now provide full learning curves in Fig. 7
> > > >
> > > >
> > > > >Theorem 4 builds on the technique in He et al.
> > > >
> > > > We have better contextualized the theorem in the main text. We have added the following text in Lines 339 in Sec. 3
> > > > Our derivation builds on the entropy-regularized half-reparameterization policy-gradient theorem for Gaussian mixture policies introduced by He et al. (2025), which provides a low-variance estimator for the standard entropy-regularized actor–critic objective with extrinsic rewards and a soft value function.
> > > >
> > > >
> > > > > References
> > > >
> > > > We have added the two references

---

### Meta-Review · Area_Chair_TmR6 · 2026-01-10

**Summary:**

The paper proposes Maxi-Mix-Mini-Com, an entropy-based objective for unsupervised action quantization to learn reusable behavioral tokens. While the motivation to bridge continuous control and LLMs via tokenization is compelling and was appreciated by reviewer kzvn, the submission suffers from some flaws in experimental design and practical utility. The decision for rejection is driven by: 1) the failure to outperform (or even compare against) simple non-learning baselines like Bang-Bang control, 2) clear evidence of non-positive transfer where pre-training hurts sample efficiency compared to learning from scratch, and 3) a structural conflict between the entropy-maximization objective and goal-reaching tasks that require termination.

**Reviewer Concerns:**

**Concerns addressed by rebuttal**
- The authors clarified the distinction between their short-lived tokens and traditional long-lived options, which satisfied Reviewer JUW5 regarding the intended scope of composability.
- Reviewers initially questioned the novelty compared to standard option discovery methods. The authors added discussions clarifying that their goal is vocabulary learning (tokenization) for interface design rather than temporal abstraction for skill discovery. This positioning was accepted by Reviewer JUW5.

**Outstanding concerns**
- Missing "Bang-Bang" baseline & unfair comparisons (Reviewer QfN9, 15JY): For a method proposing K=64 learned tokens (e.g., for Walker2d), the most direct and necessary baseline is a fixed Bang-Bang discretization (with Bernoulli policy). The authors failed to include this. The random sampling baseline used in the rebuttal appears to be an unfair comparison that artificially lowers the performance of fixed discretizations. The utility of learning tokens remains unproven against well-known fixed discretizations.
- Negative transfer & lack of utility (Figure 7): The results in Figure 7 demonstrate that the proposed pre-training method yields lower sample efficiency and final performance compared to training a continuous SAC agent from scratch. If pre-training incurs a computational cost only to hinder downstream learning (negative transfer), the method lacks practical utility. The authors failed to justify why one should use this method if it performs worse than standard RL without pre-training.
- Structural bias against goal-reaching tasks (AC Observation): The core objective (maximizing future entropy) inherently incentivizes "survival" and avoiding terminal states. This creates a conflict for goal-reaching downstream tasks where the optimal policy must terminate the episode quickly. The learned tokens will be detrimental to general task solving, and this contradicts the paper's claim of task-agnostic and general behavioral tokenization.
- Theoretical disconnect (tabular vs. continuous): The algorithm likely converges to a trivial state-independent solution in unstructured Tabular MDPs (equal action set across all states, no terminal state). Using Tabular proofs to justify a method that only works due to continuous physical constraints reveals a gap between theory and practice.

**Reviewer Scores:**

* Reviewer kzvn (8): Likely to remain 8 (Accept). This reviewer focused heavily on the novelty of the "tokenization" concept and may have overlooked the severity of the experimental flaws (negative transfer) pointed out by others.
* Reviewer JUW5 (6): Score raised from 4 to 6. The reviewer was satisfied with the conceptual clarifications. However, had they been fully aware of the negative transfer issue in Figure 7 and the missing Bang-Bang baseline, their score would likely have remained lower.
* Reviewer 15JY (4): Likely to remain 4 (Borderline Reject). They shared concerns about baselines with QfN9 and did not express a clear score raise.
* Reviewer QfN9 (2): Likely to remain 2 (Reject). This reviewer identified the performance gap in high-dimensional spaces and the lack of proper baselines. The rebuttal did not address the fundamental issue that simple discretizations might outperform the complex proposed method.

---

### Decision · Program_Chairs · 2026-01-26

Reject